# Molecular Mechanisms of the Anti-Inflammatory Effects of Epigallocatechin 3-Gallate (EGCG) in LPS-Activated BV-2 Microglia Cells

**DOI:** 10.3390/brainsci13040632

**Published:** 2023-04-07

**Authors:** Ashley Payne, Equar Taka, Getinet M. Adinew, Karam F. A. Soliman

**Affiliations:** Division of Pharmaceutical Sciences, College of Pharmacy and Pharmaceutical Sciences, Institute of Public Health (COPPS, IPH), Florida A&M University, Tallahassee, FL 32307, USA

**Keywords:** chronic neuroinflammation, BV-2 microglial cells, aging, EGCG, lipopolysaccharide (LPS), pro-inflammatory cytokines/chemokines, neuroprotection

## Abstract

Chronic neuroinflammation is associated with many neurodegenerative diseases, such as Alzheimer’s. Microglia are the brain’s primary immune cells, and when activated, they release various proinflammatory cytokines. Several natural compounds with anti-inflammatory and antioxidant properties, such as epigallocatechin 3-gallate (EGCG), may provide a promising strategy for inflammation-related neurodegenerative diseases involving activated microglia cells. The objective of the current study was to examine the molecular targets underlying the anti-inflammatory effects of EGCG in activated microglia cells. BV-2 microglia cells were grown, stimulated, and treated with EGCG. Cytotoxicity and nitric oxide (NO) production were evaluated. Immunoassay, PCR array, and WES™ Technology were utilized to evaluate inflammatory, neuroprotective modulators as well as signaling pathways involved in the mechanistic action of neuroinflammation. Our findings showed that EGCG significantly inhibited proinflammatory mediator NO production in LPS-stimulated BV-2 microglia cells. In addition, ELISA analysis revealed that EGCG significantly decreases the release of proinflammatory cytokine IL-6 while it increases the release of TNF-α. PCR array analysis showed that EGCG downregulated MIF, CCL-2, and CSF2. It also upregulated IL-3, IL-11, and TNFS10. Furthermore, the analysis of inflammatory signaling pathways showed that EGCG significantly downregulated mRNA expression of mTOR, NF-κB2, STAT1, Akt3, CCL5, and SMAD3 while significantly upregulating the expression of mRNA of Ins2, Pld2, A20/TNFAIP3, and GAB1. Additionally, EGCG reduced the relative protein expression of NF-κB2, mTOR, and Akt3. These findings suggest that EGCG may be used for its anti-inflammatory effects to prevent neurodegenerative diseases.

## 1. Introduction

Neuroinflammation is a critical factor in the etiology of neurodegenerative diseases [1]. The central nervous system (CNS) is highly susceptible to chronic inflammation, which can eventually lead to neurodegeneration, with the associated depressed ability to replenish lost or damaged neurons [2]. Neurodegenerative diseases such as Alzheimer’s and Parkinson’s Disease have been shown to arise due to the accumulation of many inflammatory mediators [2]. Alzheimer’s Disease (AD) affects 44–60 million people worldwide, with 5–7 million in the US. African Americans and Hispanics are more susceptible to cognitive impairment and delayed onset compared to Caucasians due to the prevalence of cardio/cerebrovascular disorders, genetics, socioeconomics, and racial/ethnic discrimination [3]. Current research has shown that the AD brain has increased expression of proinflammatory cytokines/chemokines, acute phase proteins, and complement elements [1]. Microglia are the local macrophages of the CNS [4] that maintain homeostasis via immunosurveillance and synaptic remodeling. Still, when triggered, they phagocytize cellular waste, expel cytokines/chemokines, and display antigens to T cells [1,5]. Chronic inflammation can exacerbate microglial activation and increase the expulsion of neuro/synaptic-toxic chemokines/cytokines [1].

Due to microglial activation, cytokines and nitric oxide play a significant role in neuroinflammation. Cytokines serve as signaling molecules that mediate cellular growth, survival, and differentiation. They also modulate leukocyte traffic, the gathering of other inflammatory factors, and immune inspection [6]. They serve to maintain microglial homeostasis through the balance of noninflammatory cytokines, i.e., Interleukin 10 (IL-10, Interleukin 13 (IL-13), and proinflammatory cytokines such as Interleukin 6 (Il-6), tumor necrosis factor-alpha (TNF-α), and Interleukin 1 beta (IL-1β), which are significantly expressed in AD. Chemokines are classified based on the primary structure or several amino acids separated by two cysteine residues, i.e., CXCL 2 (macrophage inflammatory protein 2) or Interleukin 8 (CXCL8) [7,8]. Chemokines operate in nervous system physiology by guiding neuronal migration, cell proliferation, and synaptic activity [6]. Furthermore, chemokines direct neuronal communication and may induce neuronal death via the initiation of chemokine receptors [6]. Nitric Oxide (NO) is a multivariable molecule that can be heavily expressed due to cytokine/chemokine pathways in inflammation caused by the initiation of inducible NO synthase. Continued research has shown that NO overproduction in inflammation significantly contributes to neurodegeneration in AD [9,10]. Lipopolysaccharide (LPS), a glycolipid found in Gram-negative bacteria, has been used as a microglial activator in various in vitro and in vivo model systems [11,12]. Countless research papers have demonstrated LPS use due to its ability to mimic various inflammatory effects of cytokines/chemokines, such as TNF-α and Il-6, and its specific action on the TLR4 receptor [13].

Neurodegenerative diseases such as Alzheimer’s disease (AD) are caused by immunosenescence, cellular bioenergetic dysregulation, deficient clearance mechanisms, and protein accumulation. Age-associated neurodegeneration alters these pathways’ typical function, leading to synaptic dysregulation, protein aggregation, blood–brain barrier disruption, chronic inflammation, and neuronal cell death. Nuclear Factor Kappa B (NF-κB) is a well-established transcription factor immunomodulator, aging controller, and mediator of inflammatory responses acting via proinflammatory cytokines, i.e., IL-6 and TNF-α [14,15]. The Phosphoinositide 3 Kinase/Protein Kinase B (PI3K-Akt) pathway modulates many microglial microenvironment cellular activities via its management of the mammalian mechanistic target of rapamycin (mTOR), which is a serine-threonine kinase involved in cellular metabolism, autophagy, aging, and nutrient regulation [16]. Furthermore, mTOR dysregulation has been linked to generating neurodegeneration as well as exacerbating protein accumulation (Tau protein accumulation and Amyloid beta (Aβ) in AD) [17]. Current research has led to the observation of phytochemical intervention, i.e., resveratrol and EGCG, in countering these signaling processes due to the chronic damage caused by aging and immunoregulatory insults.

On the other hand, green tea (Camellia sinensis) is consumed all over the world [18,19]. Green tea has various medicinal properties for age-related neurodegeneration, such as antioxidative, anti-viral, anti-aging, and lipid-lowering activities [20,21], due to its abundance of polyphenols or flavan-3-ols (also known as catechins). The dominant catechin in green tea is Epigallocatechin-3-gallate (EGCG), which has shown tremendous promise in neuroprotection due to its antioxidative and anti-inflammatory attributes [22,23]. Recent studies have indicated that in neurodegeneration, phytochemical intervention, i.e., resveratrol and EGCG, is caused by aging and immunoregulatory insults. EGCG’s ability to reduce microglial activation, mitochondrial dysfunction, and oxidative stress in neurodegenerative disease correlates to its unique structure and bioenergetic capabilities [24,25]. EGCG has displayed neuroprotective ability in halting Aβ formation and accumulation. Still, there is a dearth of information on the signaling mechanisms involved.

This study aimed to identify EGCG’s anti-inflammatory and antioxidant activities in an established in vitro model system (BV-2) of microglial cells. Some of the significant findings from this study are that EGCG significantly reduced the protein and mRNA expression of NF-κB, PI3K-AKT, and mTOR and reduced the production of NO, IL-6, and CCL-2. Our results suggest that EGCG may be useful in delaying the onset of inflammation and oxidative stress–mediated neurodegenerative diseases involving excessively activated microglial cells.

## 2. Materials and Methods

### 2.1. Chemical and Reagents

BV-2 microglial cells were kindly provided by Elizabeta Blasi [26]. Gibco DMEM, high-glucose GlutaMAX™ Supplement media, high-glucose HEPES, no phenol red, Gibco Penicillin–Streptomycin (10,000 U/mL), HBSS, and 10% heat-inactivated fetal bovine serum were purchased from Thermo Fisher Scientific (Waltham, MA, USA). A total of 50 mg (−)-epigallocatechin gallate ≥ 95 and Lipopolysaccharides (LPS) from *Escherichia coli* O111: B4 γ-irradiated were purchased from Sigma Aldrich (St. Louis, MO, USA). Griess Reagent was purchased from Sigma Aldrich (St. Louis, MO, USA). The individual mouse Quantikine^®^ ELISA kits for IL-6 and TNF-α were obtained from R&D Systems (Minneapolis, MN, USA). Aurum total RNA mini kit, iScript™ Advanced cDNA Synthesis Kit,2-mercaptoethanol (14.2 M) 25 mL, ≥98% pure, SsoAdvanced™ Universal SYBR^®^ Green Supermix, iScript™, Advanced cDNA Synthesis Kit, and mouse PrimePCR™ inflammation array were purchased from Bio-Rad (Hercules, CA, USA). PrimePCR™ arrays utilized in this study were mTOR, NO, PI3K-Akt, and NF-B signaling pathways and were purchased from Bio-Rad (Hercules, CA, USA). Pierce BCA Protein Assay kit was purchased from Bio-Rad (Hercules, CA, USA). Protease and phosphatase inhibitors were obtained from G-Biosciences (Saint Louis, MO, USA). In addition, the 12-230 Separation Module, 8 *×* 25 capillary cartridges, Anti-Rabbit Detection Module, HeLa Lysate Controls, and Erk1 Primary Antibody were purchased from ProteinSimple (San Jose, CA USA). Primary Antibodies NF-κB2 p100/p52, (Akt3 (ElZ3W), mTOR (7C10) Rabbit mAb), GAPDH, and Alpha-actinin, and secondary antibody Anti-rabbit IgG-HRP were purchased from Cell Signaling (Danvers, MA, USA).

### 2.2. BV-2 Cell Culturing and Treatments

Cells (BV-2) were grown in complete high-glucose media supplemented with 10% heat-inactivated FBS and 1% penicillin/streptomycin. Culture conditions were maintained at 37 °C in 5% CO_2_/atmosphere; media was replaced every 2–3 days, and cells were sub-cultured. Plating media consisted of DMEM (minus phenol red), 2.5% FBS, and no penicillin/streptomycin for experiments. The EGCG stock was prepared by dissolving 9.2 mg in 0.400 mL of deionized water and 1.6 mL of Phenol Free DMEM Media to obtain 2 mL of 10 mM of EGCG. Different concentrations were prepared from a 10 mM stock solution of EGCG for further experiments. Activation of BV-2 cells was established using 1 µg/mL of LPS.

### 2.3. Cell Viability

Cell viability was assessed using resazurin (AlamarBlue^™^, Thermo-Fisher Scientific, Waltham, MA, USA) indicator dye. Approximately 5 × 10^5^ cells/mL (100 µL per well) were plated in 96-well plates overnight. The next day, cells were first stimulated with 1 µg/mL LPS for 1 h, then treated with different concentrations of EGCG (0–350 µM) and incubated at 37 °C for 24 h. Following the desired time points, 20 µL of AlamarBlue^™^ solution (0.5 mg/mL) was added and incubated for another 4–6 h. The fluorescent signal was monitored using a 485 nm excitation wavelength and a 590 nm emission wavelength. The fluorescent and colorimetric signal generated from the assay is proportional to the number of living cells in the sample.

### 2.4. Quantification of NO

Nitric oxide (NO) production of BV-2 cells was evaluated using the Griess Reagent Assay. The Griess reagent was prepared by mixing an equal volume of 1.0% sulfanilamide in 0.5 N HCl and 0.1% N-(1-naphthyl)-ethylenediamine in deionized water. Briefly, BV-2 cells (5 × 10^4^ cells/well, in a 96-well plate) were seeded overnight. The next day, plates were prepared following the manufacturer’s recommendations (Sigma Aldrich, St Louis, MO, USA). An equal volume (50 μL) of each experimental sample (cell culture supernatant), standard, and Griess Reagent were combined, incubated for 10 min at room temperature, and protected from light. A standard curve for NO was generated from dilutions of sodium nitrite (NaNO2) (1–160 μM). Samples were analyzed at 550 nm on a UV microplate spectrophotometer (model 7600, version 5.02, Cambridge Technologies Inc., Worthington, MN, USA).

### 2.5. IL-6 and TNF-α ELISA Quantification

This experiment used R & D Systems, mouse IL-6, and a TNF-α ELISA kit to measure IL-6 and TNF-α release into BV-2 cell supernatant quantitatively. A total of 5 × 10^5^ cells/mL of BV-2 cells were seeded in 6-well plates (2 mL per well) overnight to attach. The next day, cells were treated with control (no treatment), 150 μM of EGCG only, 1 µg/mL of LPS only, and 150 μM of EGCG + 1 µg/mL of LPS, then incubated for 24 h. After 24 h exposure, the supernatant of each sample was collected into 5 mL tubes and centrifuged at 1000 rpm for 5 min at 4 °C to remove any particulate material. IL-6 and TNF-α ELISA kit reagents include the following: 96-well plate, the wash buffers, assay diluents, color reagents A & B, stop solution, conjugates, and standards. Briefly, seven serial diluted standards in pg/mL (700, 350, 175, 87.5, 43.8, 21.9, 10.9) and blank (no standard) were prepared. A total of 100 µL (in triplicate) of the standard, blank, and supernatant sample were added to each well in the plate, covered with an adhesive strip, and then incubated for 2 h at room temperature. Following this, the plate was washed; then, 100 µL of TNF-α or IL-6 conjugate was added to each well and incubated at room temperature for 2 h. The plate was washed again, and 100 µL of substrate solution was added to each well and incubated in the dark for 30 min. Finally, 100 µL of stop solution was added; then, the optical density of each well was read at 450 and 540 nm using the microplate reader (model 7600, version 5.02, Cambridge Technologies Inc.; Worthington, MN, USA). Optical errors were calculated by subtracting the absorbance of 540 nm from 450 nm. The standard curve was created by plotting the average of the triplicate optical density of each of the seven serial diluted standards. The concentration (pg/mL) of the samples was determined using the standard curve. This data obtained (concentrations of the sample [pg/mL]) were analyzed using GraphPad Prism 6 (version 6.07; Graph Pad Software Inc., San Diego, CA, USA, by one-way ANOVA with Tukey’s post hoc multiple comparisons test).

### 2.6. RNA Extraction and Reverse Transcription to cDNA

The Bio-Rad Aurum™ total RNA mini kit (Bio-Rad, Hercules, CA, USA) manufacturer’s protocol was used for RNA extraction. Briefly, the BV-2 cells were seeded (5 × 10^5^ cells/mL) in T-75 flasks (15 mL/flask) and treated for 24 h, as previously described. The cells were harvested after 24 h and then washed twice with PBS. Cell pellets were lysed with the 350 µL lysis solution and 350 µL of 70% ethanol and pipetted up and down. The homogenized lysate was put into an RNA binding column and centrifuged for 30 s. Then, 700 µL of high stringency wash solution was added to the RNA binding column and centrifuged for 30 s; 700 µL of low stringency wash solution was added to the RNA bind column after the column was replaced and centrifuged for 30 s and an additional 2 min to remove the residual wash solution (centrifugation was performed at 13,000× *g*). The RNA binding column was transferred to a 1.5 mL capped microcentrifuge tube, and 80 µL of the elution solution was added, allowing for the solution to saturate the membranes for one minute. The 1.5 mL microcentrifuge tube was centrifuged for 2 min to remove the total RNA. The eluted RNA was stored at −80 °C for later use. RNA purity and integrity were determined using Nanodrop (Thermo Fisher Scientific, Wilmington, DE, USA). The cDNA strand was formulated from the RNA using Bio-Rad iScript^™^ (Bio-Rad, Hercules, CA, USA) advanced reverse transcriptase following the manufacturer’s instructions. A solution was created from 4 µL of the 5X iScript^™^ advanced reaction mix, 1 µL of reverse transcriptase, 5 µL of the RNA sample (250 ng/5 µL) for RT-PCR assay, 9 µL of nuclease-free water, and 1 µL of the RT control assay template, for a total of 20 µL. The reverse transcription thermal cycling program included two steps: 46 °C for 20 min and then 95 °C for 5 min.

### 2.7. PrimePCR Array Analysis of Inflammatory Cytokine/Chemokines, Akt3, mTOR, NF-κB, and NO Signaling Pathways

The real-time PCR reaction was performed using the manufacturer’s protocol (Bio-Rad, Hercules, CA, USA). A total of 10 µL of the cDNA of the sample (10 ng cDNA/reaction) plus 10 µL of Ssoadvanced^™^ universal SYBR Green Supermix was added to each well of the PrimPCR™ array plate. A PCR control assay template was added to the appropriate PCR control well. The plates were then sealed, mixed on the belly dancer machine, and centrifuged for 1 min at 10,000 rpm. Following these steps, the plate was loaded into the real-time PCR machine. The thermal cycling process, including the initial hold step, was set at 95 °C for 2 min and denaturation at 95 °C for 5 s, followed by 40 cycles of 60 °C for 30 s (annealing/extension) and 75 °C for 5 s/step (melting curve) using the Bio-Rad^™^ CFX96 Real-Time System (Bio-Rad, Hercules, CA, USA). Finally, the Bio-Rad^®^ Mouse Inflammation, NF-κB, PI3k-Akt, mTOR, and NO signaling PrimePCR™ array data were analyzed via Bio-Rad CFX96 Manager software, version 3.1 (Bio-Rad, Hercules, CA, USA), which calculates fold change/regulation using the ΔΔCT (threshold cycle) method using the 2^−ΔΔCT^ formula. The *p*-values were calculated based on a student’s *t*-test of the replicate 2^−ΔCT^ values for each gene in the control group and experimental groups. PrimePCR™ array results were normalized to GAPDH using the Bio-Rad CFX96 Manager software.

### 2.8. ProteinSimple Western Analysis

#### 2.8.1. Protein Assay

Cells were treated, harvested, and centrifuged as discussed above. Then, the cell pellet was obtained and resuspended in 200 µL of lysis buffer (mixed with protease and phosphatase inhibitors) and incubated on ice for 30 min. Lysate was sonicated for 30 s using a Sonic Dimembrator (Fisher Scientific, Hampton, NC, USA) and centrifuged at 3000 RPM for 5 min. The supernatant (protein) was collected, and the protein concentration was determined using the Pierce^™^ BCA Protein Assay Kit. A series of concentration standards ranging from 0–2 mg/mL were prepared using bovine serum albumin (BSA). A total of 10 µL of each unknown protein sample and standard was used (sample to working reagent ratio 1:20). Then, 200 µL of the working reagent was added to each well of the 96-well microplates and mixed on a plate shaker for 30 s. The plate was covered and incubated at 37 °C for 30 min and left to cool at room temperature. Protein concentrations were quantified at 595 nm wavelength with the microplate reader Infinite M200 (Tecan Trading AG, Männedorf, Switzerland).

#### 2.8.2. ProteinSimple Western Assay

ProteinSimple automated WES™ analysis was used for protein quantification (ProteinSimple, San Jose, CA, USA). All reagents were provided by ProteinSimple, and the analysis was performed following the user’s manual. In brief, the first protocols were optimized for each antibody and protein loading. After optimization, the total protein concentration was 2 mg/mL for NF-κB and Akt3 and 1 mg/mL for mTOR. The protein extracts were mixed with a master mix to give a final concentration of 0.2 mg/mL total protein, 1× sample buffer, 1× fluorescent molecular weight markers, and 40 mM dithiothreitol. Samples were heated at 95 °C for 5 min. The primary antibody dilution factors used were: mTOR 1:150, Akt3 1:100, NF-κβ 1:100, GAPDH (internal control) 1:125, and Alpha-Actinin (internal control for high MW) 1:100. After samples were prepared and heated, they were loaded in the plate. Blocking solution (antibody diluent), appropriate primary antibodies, horseradish peroxidase-conjugated secondary antibodies, chemiluminescent substrate, separation and stacking matrices, and washing buffer were loaded into specified wells in a microplate. Then, the microplate was loaded into the device following the manufacturer’s instructions (ProteinSimple, San Jose, CA, USA). Target proteins were confirmed via a primary antibody and immunosorbed using an HRP-conjugated secondary antibody and chemiluminescent substrate. Chemiluminescence was obtained via a charge-coupled device camera, and the digital image was analyzed and quantified using ProteinSimple Compass software. ProteinSimple WES™ data were normalized using GAPDH (NF-κB and Akt3), whereas α-actinin was used to standardize the mTOR data.

### 2.9. Statistical Analysis

Data evaluation was performed using GraphPad Prism (version 6.07). All data were expressed as mean ± standard error from 3 independent experiments, and the significance of the difference between the groups was calculated using a one-way ANOVA, followed by Tukey’s post hoc means comparison test or Student’s *t*-test.

## 3. Results

### 3.1. EGCG Decreases Cell Viability of BV-2 Microglial Cells

EGCG has been shown to have anti-inflammatory and antioxidative properties [22,23]. To ascertain the effects of EGCG cytotoxicity, a cell viability assay was performed. Cell viability analysis showed that EGCG reduced cell survival at high concentrations. Figure 1 shows that the decreases in cell viability occurred at 175 µM and significantly decreased at 350 µM. For further inflammatory studies, 150 μM was selected for the LPS and NO studies due to cell survival being 95%.

### 3.2. EGCG Inhibits NO Production in BV-2 Microglial Cells

Nitric Oxide (NO) is a valuable indicator of oxidative stress and inflammation [27]. Colorimetric determination utilizing the Griess reagent was used to ascertain EGCG’s effects on NO production. LPS at 1 µg/mL was used to induce inflammation. The NO assay showed that LPS increased NO production, and there was a significant decrease of NO at EGCG 150 µM coupled with 1 µg/mL LPS (Figure 2). Untreated BV-2 cells were used as the negative control, whereas EGCG 150 µM alone was applied as the positive control.

### 3.3. EGCG Attenuates Proinflammatory Cytokines

Cytokines are important modulators and biomarkers for neuroinflammation [28,29]. Common proinflammatory cytokines are IL-6, TNF-α, IL-1β, and IL-10. To explore EGCG’s effects on inflammatory cytokine activity, we utilized individual ELISA analysis on TNF-α and IL-6 via the Quantikine^®^ mouse ELISA kits. The cytokine evaluation showed that concentrations of EGCG at 150 µM coupled with 1 µg/mL LPS significantly attenuated IL-6 compared to LPS (1 µg/mL) (Figure 3). In Figure 4, interestingly, EGCG 150 µM + 1 µM LPS increased TNF-α production. LPS (1 µg/mL) also elevated TNF-α. Untreated BV-2 cells were used as the negative control, and EGCG 150 µM alone was applied as the positive control. Appendix A details the findings related to EGCG effects for this experiment.

### 3.4. EGCG Acts to Downregulate Proinflammatory Cytokines and Upregulate Autophagic Neuroprotective Genes

As mentioned, cytokines and chemokines act as regulators of the innate neuroimmunological response in the microglia. The PrimePCR™ array was used to further validate the ELISA data and to investigate other proinflammatory cytokines affected by EGCG. Our results demonstrated that EGCG 150 µM coupled with 1 µg/mL LPS elevated the gene expression of Interleukin 3 (IL-3), Interleukin 11 (IL-11), and Granulocyte-macrophage colony-stimulating factor (GM-CSF or CSF2), as shown in Figure 5. EGCG 150 µM coupled with 1 µg/mL LPS was able to reduce the gene activity of macrophage migration inhibitory factor (MIF), as displayed in Figure 6. Additionally, Figure 6 shows EGCG 150 µM with 1 µg/mL LPS attenuated Chemokine C-C motif ligand 2 (CCL2) aka Monocyte chemoattractant protein 1 (MCP-1), and Tumor necrosis factor (ligand) superfamily, member 10 (TNFs10). The control was untreated BV-2 cells and varied in expression due to EGCG’s effects on the specific mRNA expression of the gene. Appendix A details the inflammatory cytokine function and fold change.

### 3.5. EGCG Modulates Neuroimmunomodulation of Inflammation via NF-κB, PI3k-Akt-mTOR, and NO Pathway

Inflammatory signaling pathways have been shown to become dysregulated and maladjusted, leading to destabilized cellular metabolism, defective clearance mechanisms, and heightened microglial stress response due to neurodegenerative disorder [30,31]. PrimePCR™ array evaluation of EGCG effects displayed the expression of many genes; Appendix A lists in more detail all of the genes that appeared in this analysis. Figure 7, Figure 8, Figure 9, Figure 10, Figure 11, Figure 12, Figure 13 and Figure 14 show only the statistically significant genes for the aforementioned inflammatory signaling pathways. EGCG upregulated neuroprotective NF-κB genes (Figure 7), which were as follows: Colony stimulating factor 3 (CSF3), Toll-like receptor 4 (Tlr4), Toll-interleukin 1 receptor (TIR) domain-containing adaptor protein (TIRAP), Toll-like receptors (TLR1, 3 and 4), Tumor necrosis factor (TNF), heme oxygenase 1 (HMOX1), and ILR1 (Interleukin Receptor 1). EGCG downregulated proinflammatory NF-κB genes, as shown in Figure 8, which are: C-C-Ligand 5 (CCL5), Fos Proto-Oncogene, AP-1 Transcription Factor Subunit (FOS), Interleukin 1 Beta (IL1B), Mothers against decapentaplegic homolog 3 aka SMAD Family Member 3 (SMAD3), TNF receptor-associated factors (TRAF2.3 and 5), RelB Proto-Oncogene, NF-KB Subunit (RelB), Signal Transducer and Activator of Transcription 1 (STAT1), V-RAF-leukemia viral oncogene 1 (RAF1), Nuclear Factor Kappa B p-100 subunit 2 (NF-κB2), and Mitogen-Activated Protein Kinase Kinase Kinase 1 (MAP3K1). EGCG upregulated immunosurveillance PI3K-AKT genes, as seen in Figure 9, which shows Eukaryotic Translation Initiation Factor 4E (ELF4E) and Toll-Like Receptor 4 (TLR4). Figure 10 shows that EGCG downregulated proinflammatory PI3K-AKT genes, which are as follows: Son of sevenless homolog 1 (Drosophila) (SOS1), Raf proto-oncogene serine/threonine-protein kinase aka proto-oncogene c-RAF (RAF1), Beta-Glucuronidase (GUSB), Insulin-Like Growth Factor I receptor (IGF1R), Cyclin-dependent kinase inhibitor 1B (CDKN1B), Growth Factor Receptor-Bound Protein 2 (GRB2), Mature T Cell Proliferation 1 (MTCP1), and Thymoma Viral Proto-Oncogene 3 (AKT3). Figure 11 displays EGCG downregulation of proinflammatory mTOR genes, which were: Regulatory Associated Protein of mTOR Complex 1 (RPTOR), Mitogen-Activated Protein kinase kinase (MAPK3), Thymoma Viral Proto-Oncogene 3 (AKT3), Protein Kinase Adenosine Monophosphate Activated Non-Catalytic Subunit 2 (PRKAB2), and mTOR. EGCG upregulated insulin signaling linked mTOR genes, as displayed in Figure 12, which are: Insulin2 (INS2) and Phospholipase D2 (PLD2). Figure 13 shows EGCG downregulation of oxidative stress–producing NO genes, which are: Glutathione Peroxidases 1 and 4 (GPX1 and 4), Growth Arrest and DNA Damage Inducible Protein (GADD45A), Nitric Oxide Synthase 1 (NOS1), Cathepsin B (CTSB), Cytochrome B-245 Alpha Chain (CYBA), Immunity-related GTPase family M protein aka Interferon-Inducible Protein 1 (IRGM1), Proliferation and Apoptosis Adaptor Protein 15A (PEA15A), and Hepsin (HPN). Finally, EGCG upregulated the bioenergetic regulator GRB2-associated binding protein 1 (Gab1). For this experiment, nontreated cells were used as the negative control, and EGCG 150 µM alone was utilized as the positive control. Variations of controls are due to the effects of EGCG on particular mRNA-expressed genes. Appendix A gives further information related to each gene and fold change.

### 3.6. EGCG Promotes Neuroprotection by Diminishing Protein Levels of NF-κB2, AKT3, and mTOR

Inflammatory signaling regulation at the protein level allows for the proper regulation of cellular metabolic and proliferative processes. Western blot validates the inflammatory signaling pathway action of inflammatory response and biomarkers. EGCG was shown to decrease the protein expression of NF-κB2 (Figure 15A,B and Appendix A), Akt3 (Figure 16A,B and Appendix A), and mTOR (Figure 17A,B and Appendix A).

## 4. Discussion

Microglia are the immunomodulator cells of the central nervous system [5]. The activation of these cells has been linked to various neurodegenerative diseases, i.e., AD. The medicinal properties of polyphenols in the human diet are still being investigated, especially in everyday foods such as peanuts, curcumin, the skin of red grapes, and green tea [32,33,34]. New studies involving EGCG have revealed comparative findings in cell survival, inflammatory cytokines, chemokine activity, and inflammatory regulation [35,36]. For this study, the fluorometric/colorimetric (resazurin) assay was used to assess the cytotoxicity of EGCG based on its validation of metabolic integrity, rather than the MTT(3-(4,5-dimethylthiazol-2-yl)-2-5-diphenyltetrazolium bromide) assay, which assesses cellular mitochondrial membrane potential [37,38]. BV-2 and N9 cells exhibit similar functions in primary cells and are most suited for in vitro neuroinflammatory/degenerative study [39,40,41,42,43].

NO has been used as a signal of inflammation and oxidative stress in vivo [10]. Neuroinflammation can increase the amount of NO in microglia, enhancing neurodegeneration [44]. In this study, we showed that the 150 µM concentration of EGCG significantly reduced NO (linked to aging and neuroinflammation). EGCG’s anti-inflammatory and antioxidative functions may regulate iNOS expression to reduce nitric oxide, which will diminish cellular stress and neuroinflammation. A previous work [45] showed that EGCG could halt the induction of nitric oxide synthase via the downregulation of LPS-induced action via NF-kB.

Inflammatory cytokines and chemokines are significant indicators of neuroinflammation. The results of our ELISA experiments demonstrated that EGCG might decrease IL-6 while increasing TNF-α. IL-6 is an essential proinflammatory regulator that acts on JAK/STAT pathways in regulating inflammation. The reduction of IL-6 may be correlated to its regulation by nitric oxide in epithelial cells [46]. It has been shown that IL-6 is linked to the activation of NF-kB and TLR receptors, which leads to neuroinflammation. TNF-α has been shown to have pro- and anti-inflammatory properties and is also involved in a wide range of cellular survival and regulatory processes. To explain the observed effects of TNF-α exhibited by EGCG 150 µM combined with LPS, we propose the following: (1) TNF-α may serve to enhance the clearance mechanisms that are involved in chronic inflammation via its actions on TNFRII receptors to stimulate macrophages or monocytes to reduce the inflammatory response [47]. (2) The phytoestrogen properties of EGCG may be responsible for the increasing levels of TNF-α by acting on estrogen receptors in microglia [48,49]. (3) It has been shown that EGCG affects nuclear factor-erythroid factor 2-related factor 2 (Nrf2), which may have an impact on the expression of TNF-α [50,51]. (4) EGCG has been shown to enhance TNF-α associated apoptosis in rheumatoid arthritis (RA) synovial fibroblasts [52]. Granulocyte Colony Stimulating Factor (CSF2) acts on the GM-CSF receptor to modulate inflammation [53,54]. Although it is shown to be a pro-inflammatory agent in the literature, it may have neuroprotective properties [55]. The upregulation of CSF2 by EGCG may heighten the microglia’s immunosurveillance properties to allow for a neuro-defensive mechanism for homeostasis in the inflamed microenvironment. Related research has shown that EGCG increased G-CSF and neutrophilia to reduce sepsis [56].

The PrimePCR™ array evaluation of mTOR revealed that EGCG downregulated mTOR, which may act as an important mediator of various medicinal functions, i.e., anti-aging, autophagy regulation, neuroprotection, and glucose and bioenergetic mechanisms [57,58]. mTOR signaling analysis demonstrated EGCG might be acting to inhibit PI3k-Akt (EGCG downregulation of Akt3), which would downregulate mTORC1 (reduced mRNA mTOR expression), thus increasing autophagy. mTOR is a nutrient-sensing pathway and inhibits autophagy via the Ulk/Atg12/p100 complex [58]. EGCG may regulate this nutrient-sensing capability by mimicking a starved state (downregulation of mTOR), which could lessen chronic inflammation. Another possible route is EGCG regulation of glucose homeostasis by either the insulin pathway (as evidenced by the upregulation of Ins2 and PLD2 as well as downregulation of IGF1R) by promoting more insulin generation or via the TSC1/TSC2 complex, primarily TSC2 (tuberin) to inhibit mTORC1 [59,60]. Further investigation into EGCG’s effects on amino acid metabolism and AMPK signaling (downregulation of PRKAB2) is needed to ascertain their involvement in cellular energetics in mTOR regulation. Our research further showed that our results were similar to previous research on EGCG’s management of MAPK signaling exemplified by the PI3k-AKT downregulation of GRB2, SOS1, RAF1, and MAP3K1 (also shown in the NF-κB and mTOR results). Ongoing research has focused on the role of MAPK and insulin in glucose metabolic dysregulation in generating neurodegeneration via instigating metabolic stress [61,62,63].

The link between PI3k-Akt and NF-κB was established with the upregulation of TLR4 and the downregulation of FOS (a transcription factor that can modulate apoptosis via AP-1). In addition, the PI3K-AKT pathway in our research outcomes displayed the downregulation of CDK1nb, which is associated with cell cycle arrest or cellular senescence. The inhibitory action of EGCG on the NF-κB pathway as well as its actions to downregulate TRAFs (TRAFs 2, 3, and 5), allows for TLR to mediate TLR signaling (upregulation of TLRs 1, 3 and 4), thus reducing inflammatory cytokine production (CCL5 and IL-1b). EGCG also reduced bcl3 expression, which is involved in directing NF-κB activity [64] and lipid metabolism [65]. NF-κB analysis demonstrated SMAD3 upregulation, thus showing a connection to transforming growth factor beta (TGF-β), which may be another clearance mechanism utilized by EGCG in mediating aging and neurodegeneration. EGCG inhibition of NF-κB also promotes the inhibition of aberrant NO production characterized in neuroinflammation, as evidenced in the downregulation of NOS1 (which is implicated in vascular inflammation and is connected to the NF-kB pathway). HMOX1 upregulation demonstrates a connection to oxidative stress regulation by EGCG.

Regarding the NO signaling pathway, our results show that EGCG quells overactive glutathione peroxidases (GPX1 and GPX4) or the overproduction of SOD by CYBA to reduce ROS. Nitric oxide has been shown to regulate ferroptosis, which may be a possible treatment for neurodegeneration [66,67,68,69,70]. Nitric oxide signaling inquiry showed downregulation of Hepsin, for which there is a dearth of information related to chronic neuroinflammation and aging microglia. A role for protein aggregation control by EGCG is displayed by the decreased mRNA expression of PRKCAB2, CTSB, and GUSB, which shows a possible role in its ability to reduce protein aggregation and misfolding associated with neurodegeneration. Our results demonstrated a potential role of senescent cellular modulation of aging in microglia and an autophagic role by the downregulated expression of Pea15a, GADD45a, FOS, CTSB, and CDKN1B evaluation. In this work, we were limited to utilizing the BV-2 microglial cell model and were unable to study the connection of astrocytes and neighboring glial microenvironment regulators involved in neuroinflammation and immunosenescence.

The genes chosen for Protein Simple WES™ evaluation were indicative of EGCG effects on the signaling pathways themselves, i.e., NF-κB2, mTOR, and Akt3. The inhibition of NF-κB2 by EGCG may elicit further study on the effects of NF-κB in promoting inflammation and its role in mediating aging [71]. AKT3 is one of the isoforms of the PI3K-Akt signaling pathway; it is found within the neurons and comprises 50% of the total mammalian brain [72]. This AKT isoform is associated with a dearth of research related to microglia and neurodegeneration. A role may exist in macrophage regulation [73]. Polytarchou et al., 2020 [74] demonstrated that AKT3 could generate oxidative stress and DNA breakdown by stimulating NADPH oxidase through the phosphorylation of p47^phox^ using an in vitro murine model system. Further investigation of AKT3 is its ability to modulate mitochondria and autophagy has been reported [75,76]. Most importantly, AKT3 has been shown to be involved in lysosomal dysregulation caused by cellular senescence [77]. Dubois et al., 2019 [72] showed that AKT3 modulates protection against demyelinating inflammatory disorder. Nutraceutical intervention of the PI3K-AKT pathway in microglial regulation may be promising to consider [78]. EGCG downregulation of AKT3 may utilize an anti-aging function that mediates AKT3 activity.

mTOR is a multifaceted kinase that regulates autophagy, cellular senescence, aging, and neuroinflammation [79,80,81]. mTOR also is involved in autophagy regulation [82]. More importantly, mTOR is a possible treatment for AD pathogenesis [17]. EGCG displayed a reduction of mTOR in western blot analysis. EGCG action, similar to other nutraceuticals, may be beneficial to mitochondrial bioenergetics via mTOR signaling intervention [83,84].

NF-κB2 is the gene that encodes for the NF-κB family of proteins. NF-κB is an extensively studied inflammatory pathway. As previously mentioned, NF-κB regulates NO, which mediates oxidative stress and microglial activation via TLR signaling [85]. Some less studied elements of NF-κB action correlated to AD are aging control [86], estrogenic regulative action [87], and flavonoid neuroprotection [88]. WES technology showed that EGCG downregulated NF-κB, which shows a flavonoid intervention in regulating inflammatory action. More research is necessary to understand the roles of mTOR, NF-kB, and Akt signaling in mediating the lipid metabolic effects on aging and neuroinflammation [89,90]. Research mechanisms of Tau protein contributions to aggregation were previously discussed in our prior work [91], but their relationship to microglial inflammation [92] and signaling mechanisms has yet to be determined. Our research shows that EGCG may be able to quell microglial inflammation and act as an anti-aging preventive measure in early-stage neurodegeneration. A summary of the potential mechanisms of EGCG effects is displayed in Figure 18.

## 5. Conclusions

Our research showed that EGCG shows a profound mechanism of regulating PI3K-AKT, mTOR, and NO signaling in order to decrease neuroinflammation. EGCG’s anti-inflammatory and neuroprotective attributes were displayed by its ability to diminish the action of well-established inflammatory, stimulating cytokines, i.e., IL-6 and MIF. The neuroprotective capabilities were exhibited by the upregulation of IL-3 and IL-11. EGCG displayed some autophagic properties by acting on CTSB and PRKAB2. Although LPS was utilized to stimulate the BV-2 microglial cells, EGCG demonstrated the ability to suppress microglial initiation, which is a pivotal event in neurodegenerative disease.

## Figures and Tables

**Figure 1 brainsci-13-00632-f001:**
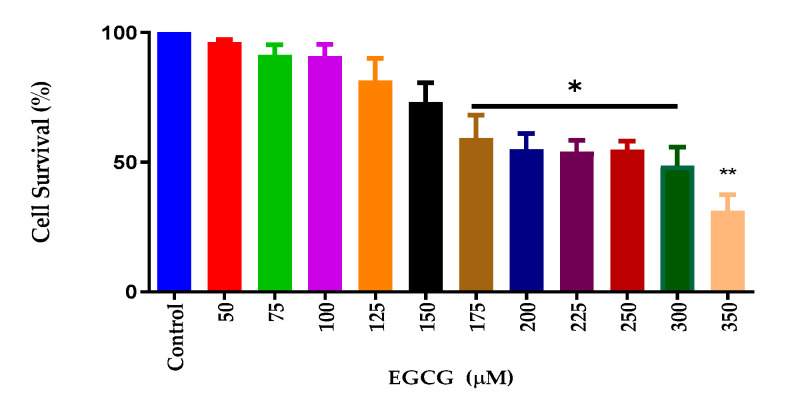
EGCG caused a concentration-dependent decrease in cell viability of BV-2 for concentrations ≥ 175 µM. AlamarBlue™ assay was used to assess cell viability. The *X*-axis represents the different concentrations of EGCG, and the *Y*-axis represents cell survival (%). As shown in the above graph, EGCG caused a concentration-dependent decrease in cell viability of BV-2 for concentrations ≥ 175 µM. EGCG 150 µM was chosen as the concentration for the remaining experiments. Values represent the mean ± SD, * *p* ≤ 0.05, ** *p* ≤ 0.01.

**Figure 2 brainsci-13-00632-f002:**
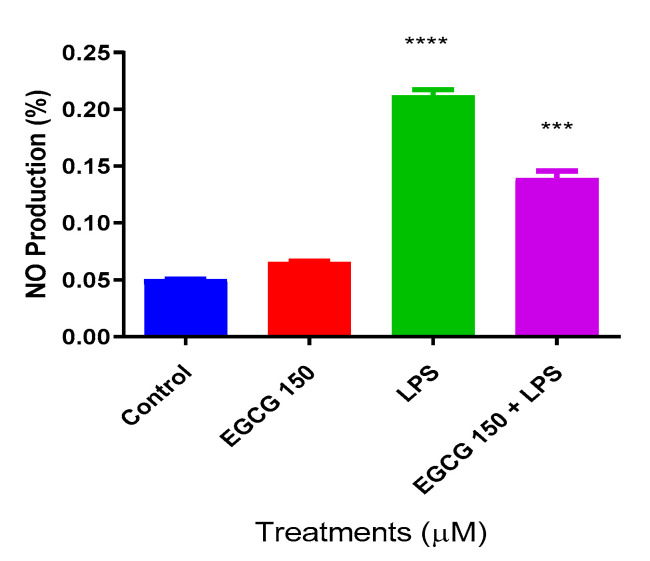
EGCG caused a concentration-dependent decrease in nitric oxide (NO) production. Griess reagent assay was used to measure nitric oxide (NO) production. The *X*-axis represents the different treatment conditions for BV-2 cells, while the *Y*-axis shows NO production (%). EGCG 150 µM showed no significant change when compared to the control. EGCG 150 µM paired with 1 µg/mL of LPS significantly reduced NO generation compared to LPS. LPS (1 µg/mL) showed a 45% increase in NO production when compared to the controls (EGCG 150 µM alone and no treatment control), whereas the EGCG 150 µM with 1 µg/mL LPS displayed a 25% decrease in NO generation. Values represent the mean ± SD, *** *p* ≤ 0.001, and **** *p* ≤ 0.0001 vs. LPS.

**Figure 3 brainsci-13-00632-f003:**
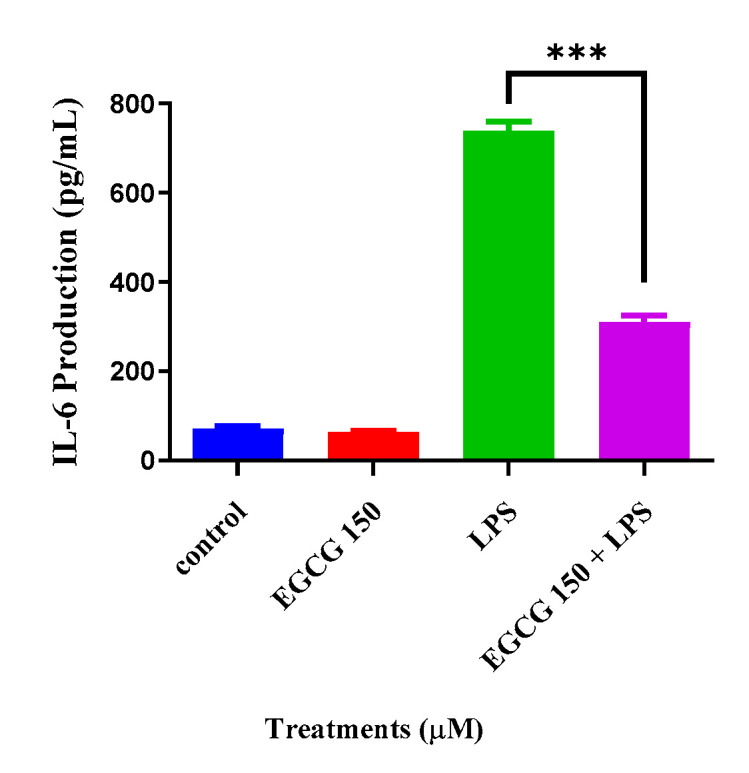
EGCG concentrations at 150 µM coupled with 1 µg/mL LPS reduce IL-6. LPS 1 µg/mL showed a 50% increase in IL-6 production when compared to both controls (untreated cells and EGCG 150 µM alone). EGCG 150 µM with 1 µg/mL LPS showed a 25% decrease in IL-6 production when compared with LPS (1 µg/mL). Values represent the mean ± SD, *** *p* ≤ 0.001 vs. LPS.

**Figure 4 brainsci-13-00632-f004:**
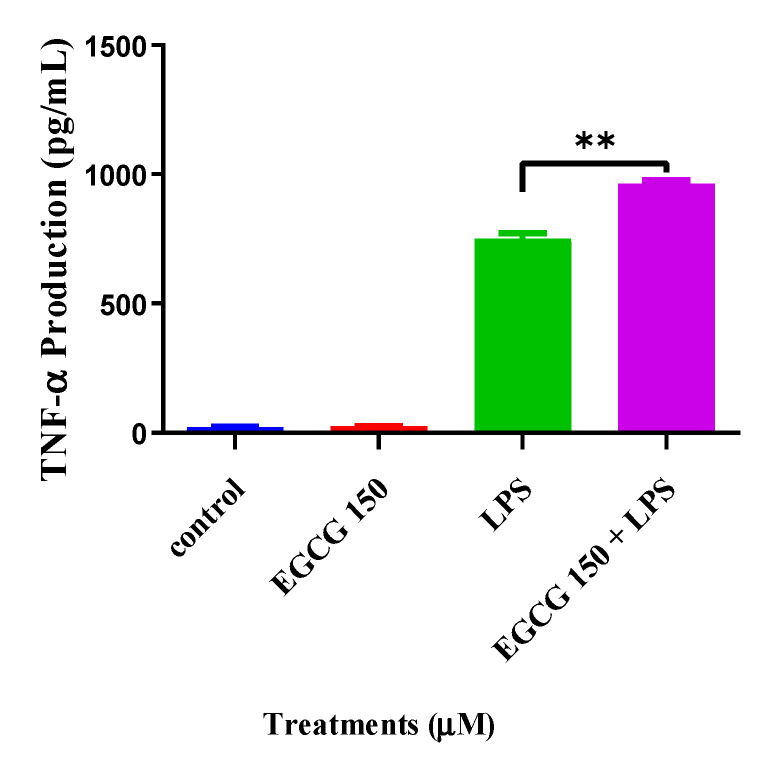
EGCG concentrations at 150 µM coupled with 1 µg/mL LPS increase TNF-α. Individual ELISA analysis of BV-2 cells shows that EGCG at 150 µM and LPS (1 µg/mL) had a 2% increase of TNF-α compared to LPS (1 µg/mL). LPS 1 µg/mL increased TNF-α by 30%. EGCG 150 µM alone showed no significant change and was similar to the control (untreated cells). Values represent the mean ± SD, ** *p* ≤ 0.01 vs. LPS.

**Figure 5 brainsci-13-00632-f005:**
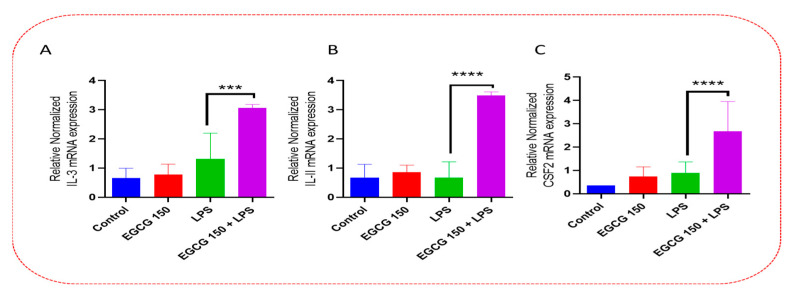
EGCG upregulates IL-3, IL-11, and CSF2. PCR array analysis of BV-2 cells shows that EGCG 150 µM + LPS (1 µg/mL) significantly increased mRNA expression of IL-3 (**A**), IL-11 (**B**), and CSF2 (**C**) compared to LPS (1 µg/mL). LPS (1 µg/mL) showed a non-significant increase when compared to EGCG 150 µM with LPS (1 µg/mL). The controls (untreated cells and EGCG 150 µM alone) displayed reduced mRNA expression when compared to LPS and EGCG 150 µM with LPS (1 µg/mL). Values represent the mean ± SD, *** *p* ≤ 0.001, and **** *p* ≤ 0.0001 vs. LPS.

**Figure 6 brainsci-13-00632-f006:**
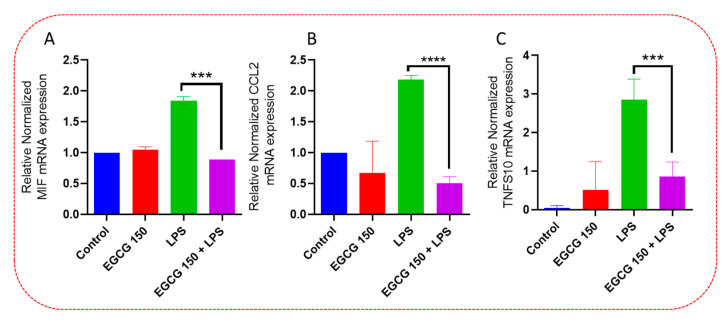
EGCG downregulates MIF, CCL2, and TNFS10. PCR array evaluation of BV-2 cells demonstrated that EGCG 150 µM + LPS reduces the mRNA expression of proinflammatory mediators: MIF (**A**), CCL2 (**B**), and TNFS10 (**C**). LPS 1 µg/mL considerably elevated mRNA expression of MIF, CCL2, and TNFS10 when contrasted to 150 µM and 1 µg/mL LPS. The controls (untreated cells and EGCG 150 µM alone) varied in expression due to the effects of EGCG on these specific genes. Values represent the mean ± SD, *** *p* ≤ 0.001, and **** *p* ≤ 0.0001 vs. LPS.

**Figure 7 brainsci-13-00632-f007:**
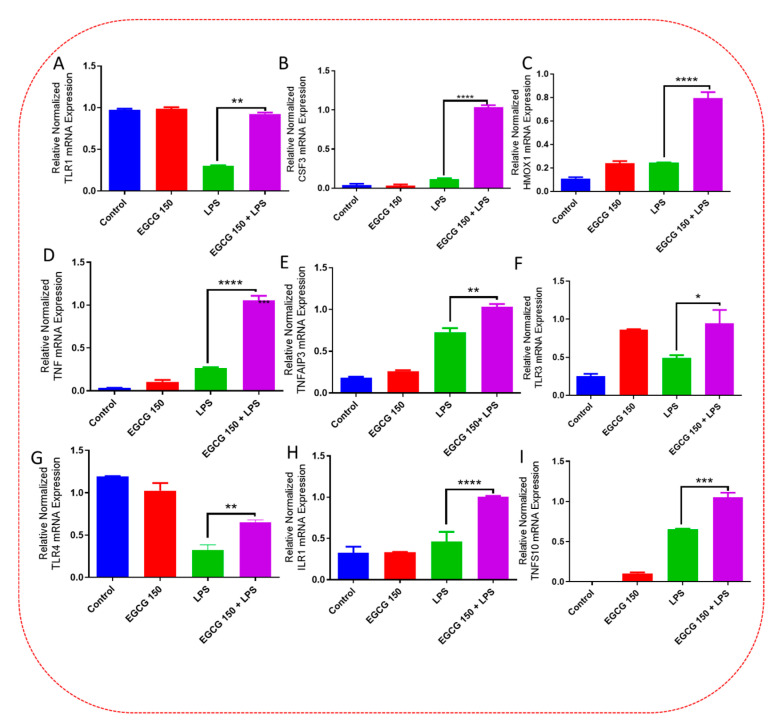
EGCG upregulated neuroprotective-regulating NF-κB genes. PCR array analysis showed that EGCG 150 µM coupled with LPS (1 µg/mL) elevated mRNA expression of the genes TLR1 (**A**), CSF3 (**B**), HMOX1 (**C**), TNF (**D**), TNFAIP3 (**E**), TLR3 (**F**), TLR4 (**G**), ILR1 (**H**), and TNSF10 (**I**) compared to LPS. Values represent the mean ± SD, * *p* ≤ 0.05, ** *p* ≤ 0.01, *** *p* ≤ 0.001 and **** *p* ≤ 0.0001 vs. LPS.

**Figure 8 brainsci-13-00632-f008:**
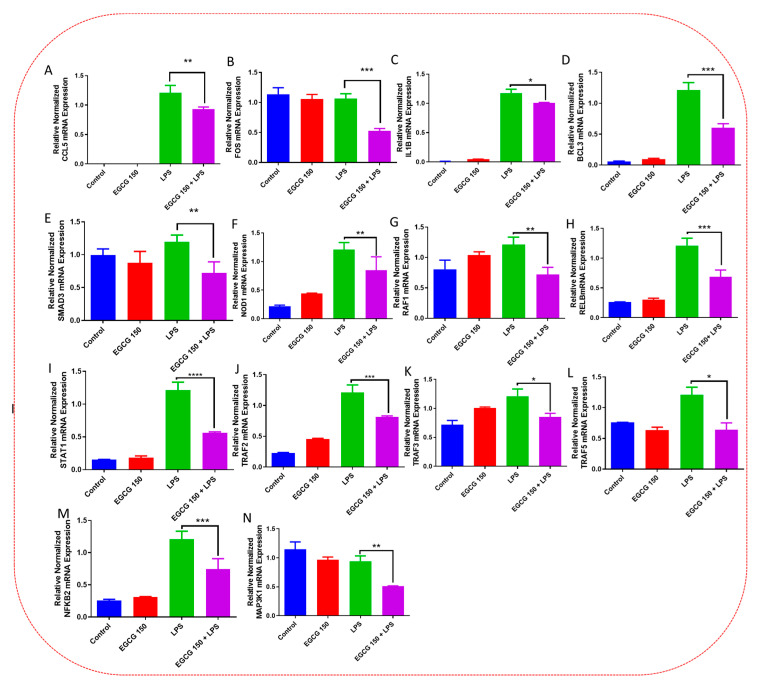
EGCG downregulated inflammation-mediating NF-κB genes. PCR array evaluation established that EGCG 150 µM coupled with LPS (1 µg/mL) curtailed mRNA expression of the genes CCL5 (**A**), FOS (**B**), IL1B (**C**), BCL3 (**D**), SMAD3 (**E**), NOD1 (**F**), RAF1 (**G**), RELB (**H**), STAT1 (**I**), TRAF2 (**J**), TRAF3 (**K**), TRAF5 (**L**), NFKΒ2 (**M**), and MAP3K1 (**N**) compared to LPS. Values represent the mean ± SD, * *p* ≤ 0.05, ** *p* ≤ 0.01, *** *p* ≤ 0.001 and **** *p* ≤ 0.0001 vs. LPS.

**Figure 9 brainsci-13-00632-f009:**
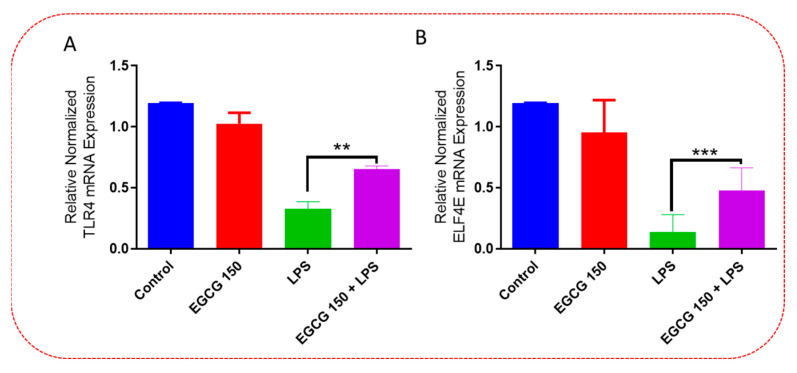
EGCG upregulated immunosurveillance PI3k-AKT genes. PCR array investigation demonstrated that EGCG 150 µM coupled with LPS (1 µg/mL) raised mRNA expression of TLR4 (**A**) and ELF4E (**B**) compared to LPS. Values represent the mean ± SD, ** *p* ≤ 0.01, and *** *p* ≤ 0.001 vs. LPS.

**Figure 10 brainsci-13-00632-f010:**
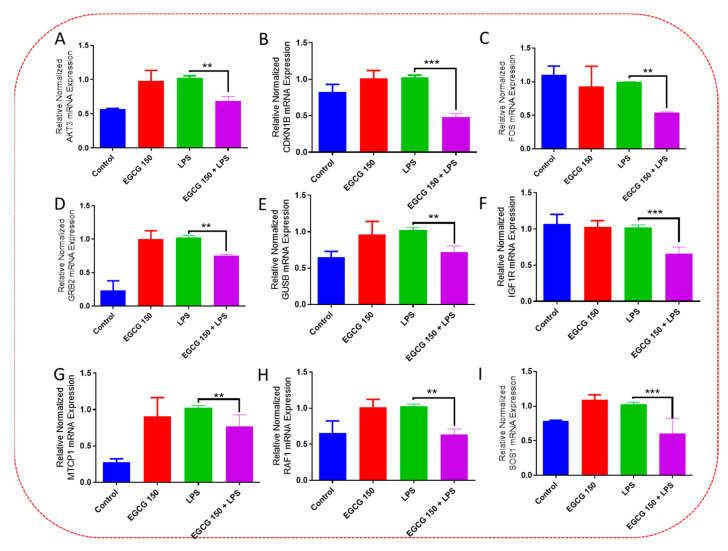
EGCG downregulated inflammatory-associated PI3k-AKT genes. PCR array inquiry displayed EGCG 150 µM coupled with LPS (1 µg/mL) decreased mRNA expression of the genes AKT3 (**A**), CDKN1B (**B**), FOS (**C**), GRB2 (**D**), GUSB (**E**), IGF1R (**F**), MTCP1 (**G**), RAF1 (**H**), and SOS1 (**I**) compared to LPS. Values represent the mean ± SD, ** *p* ≤ 0.01, and *** *p* ≤ 0.001 vs. LPS.

**Figure 11 brainsci-13-00632-f011:**
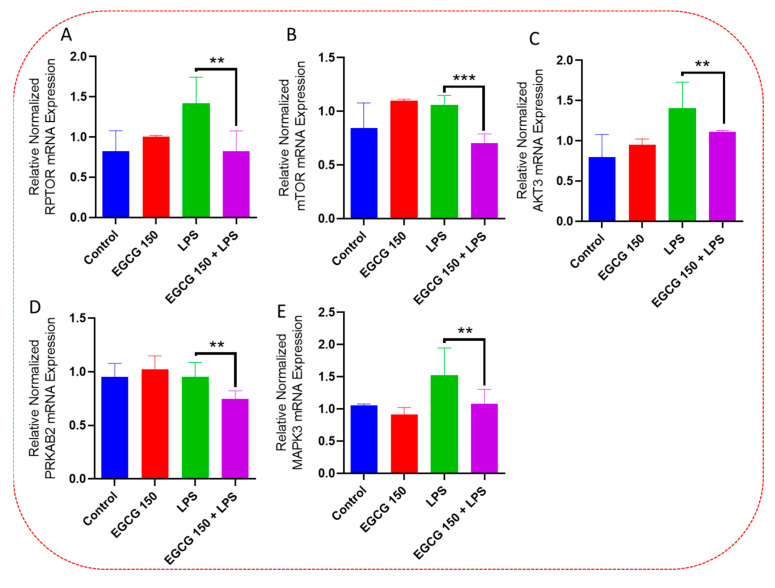
EGCG downregulated inflammatory microglial mTOR Genes. PCR array examination showed that EGCG 150 µM paired with LPS (1 µg/mL) reduced mRNA expression of the genes RPTOR (**A**), Mtor (**B**), AKT3 (**C**), PRKAB2 (**D**), and MAPK3 (**E**) compared to LPS. Values represent the mean ± SD, ** *p* ≤ 0.01, and *** *p* ≤ 0.001 vs. LPS.

**Figure 12 brainsci-13-00632-f012:**
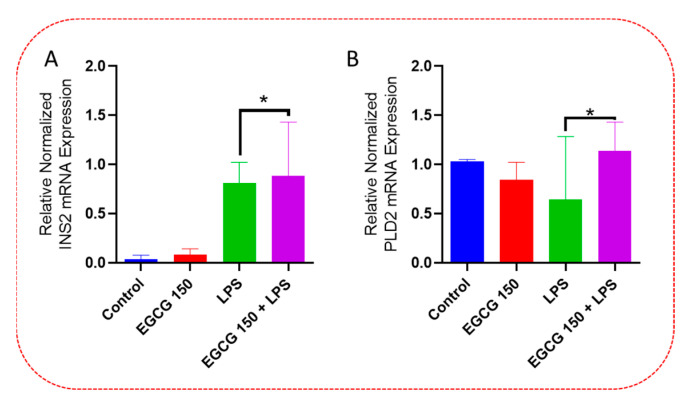
EGCG upregulated insulin signaling-related mTOR genes. PCR array experimentation indicated that EGCG 150 µM coupled with LPS (1 µg/mL) heightened mRNA expression of INS2 (**A**) and PLD2 (**B**) compared to LPS. Values represent the mean ± SD, * *p* ≤ 0.05 vs. LPS.

**Figure 13 brainsci-13-00632-f013:**
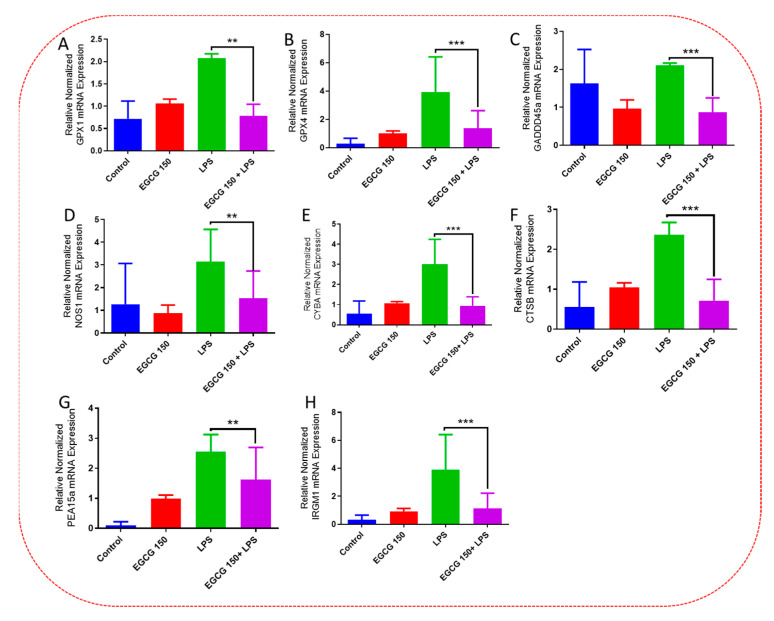
EGCG downregulated oxidative stress–producing NO genes. PCR array analysis displayed that EGCG 150 µM coupled with LPS (1 µg/mL) lowered mRNA expression of the genes GPX1 (**A**), GPX4 (**B**), GADDD45A (**C**), NOS1 (**D**), CYBA (**E**), CTSB (**F**), PEA15A (**G**), and IRGM1 (**H**) compared to LPS. Values represent the mean ± SD ** *p* ≤ 0.01, and *** *p* ≤ 0.001 vs. LPS.

**Figure 14 brainsci-13-00632-f014:**
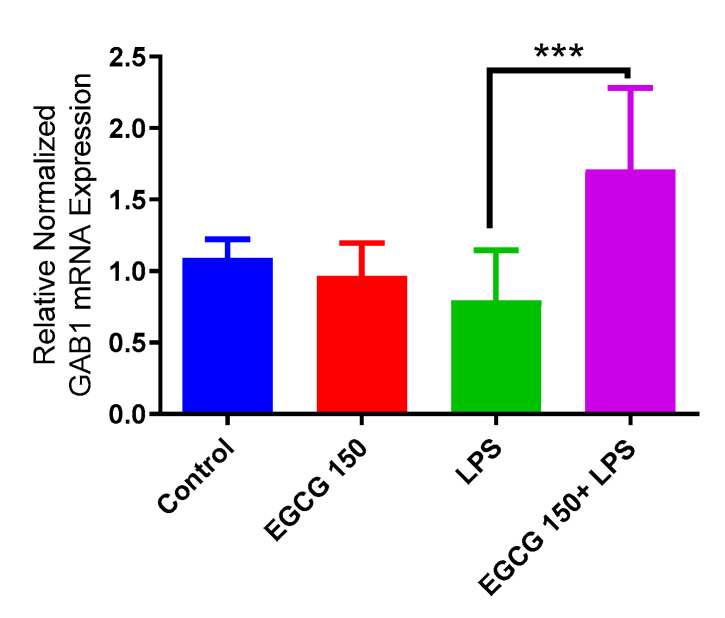
EGCG upregulated *GAB1*. PCR array investigation indicated that EGCG 150 µM coupled with LPS (1 µg/mL) increased mRNA expression of GAB1 compared to LPS. Values represent the mean ± SD, *** *p* ≤ 0.001 vs. LPS.

**Figure 15 brainsci-13-00632-f015:**
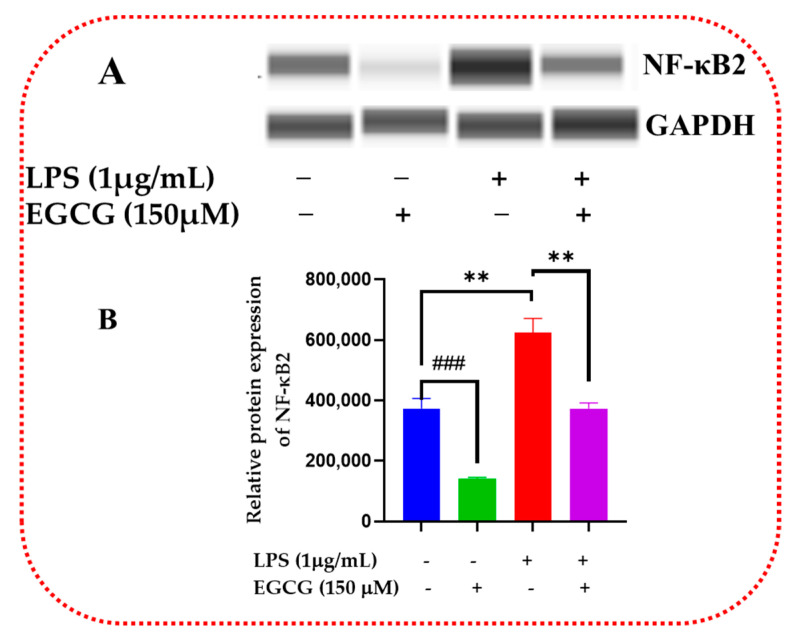
EGCG downregulated relative protein expression of NF-κB2. Western blot representative image of NF-κB2 evaluation demonstrated that EGCG 150 µM coupled with LPS 1 µg/mL displayed in bands in (**A**) and the graph in (**B**), reduced protein expression of the analyte as mentioned above compared to LPS. EGCG 150 µM alone reduced NF-κB2 protein expression. GAPDH was used as a loading control. Values represent the mean ± SD, ** *p* ≤ 0.01, vs. LPS, ^###^
*p* < 0.001 vs. control.

**Figure 16 brainsci-13-00632-f016:**
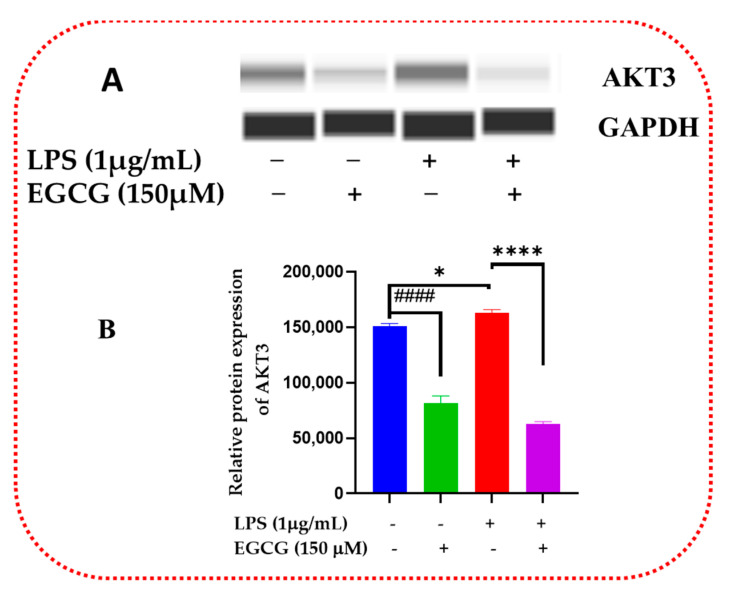
EGCG decreased the relative protein expression of Akt3. Western blot representative image of AKT3 shows that EGCG 150 µM coupled with LPS 1 µg/mL displayed in bands in (**A**) and graph in (**B**) reduced protein expression of the aforementioned analyte compared to LPS (red). EGCG 150 µM alone also decreased NF-κB2 protein expression. GAPDH was used as a loading control. Values represent the mean ± SD, * *p* ≤ 0.05, and **** *p* ≤ 0.0001 vs. LPS, ^####^
*p* < 0.0001 vs. control.

**Figure 17 brainsci-13-00632-f017:**
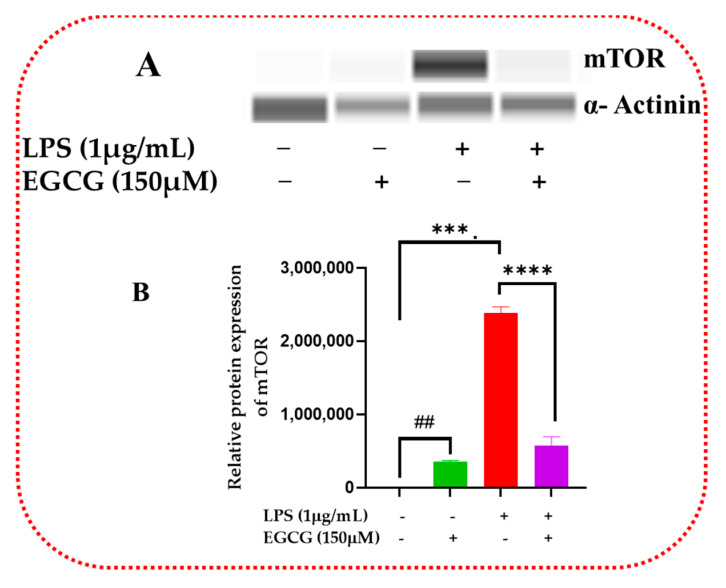
EGCG attenuated relative protein expression of mTOR. Western blot representative image of mTOR assessment displayed that EGCG 150 µM coupled with LPS 1 µg/mL, shown in bands in (**A**) and graph in (**B**), reduced protein expression of the analyte as mentioned above compared to LPS. EGCG 150 µM showed a decrease of mTOR protein expression. α-Actinin was used as loading control. Values represent the mean ± SD, *** *p* ≤ 0.001, and **** *p* ≤ 0.0001 vs. LPS, ^##^
*p* < 0.01 vs. control.

**Figure 18 brainsci-13-00632-f018:**
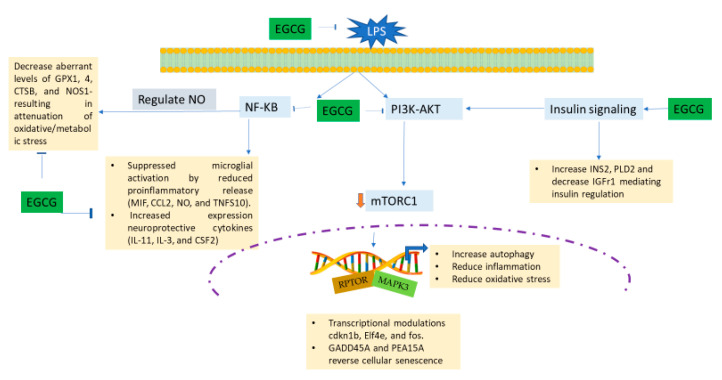
Schematic diagram showing a possible mechanism of EGCG inhibition of LPS-induced inflammatory and oxidative stress response leading to depression of neuroinflammation. Experimentation showed that EGCG diminished LPS-activated BV-2 microglial cells via the following suggested mechanisms: EGCG halted the NF-κB and the PI3K-AKT pathway, which downregulates mTORC1, thus causing an increase in its anti-inflammatory and neuroprotective attributes. EGCG also controls insulin signaling to mediate PI3k-AKT. EGCG directs the regulation of nitric oxide via NF-κB signaling, which results in the reduction of oxidative stress/metabolic stress. Additionally, transcriptional modulators may be involved in further neuro-rescue and anti-senescent capabilities of EGCG.

## Data Availability

All data generated or analyzed during this study are included in this published article.

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
