# Peer review of "Molecular Mechanisms of the Anti-Inflammatory Effects of Epigallocatechin 3-Gallate (EGCG) in LPS-Activated BV-2 Microglia Cells"

_brainsci, 2023, doi:10.3390/brainsci13040632_

Round 1

Reviewer 1 Report

The subject of this study is extremely interesting, considering that neurodegenerative diseases have a significant weight worldwide, and slowing down or stopping the evolution is often impossible.

Anti-inflammatory effect of Green Tea polyphenol Epigallocatechin-3-Gallate (EGCG) has been the subject of many studies, but the approach in this study is extremely complex. The methodology is extremely rigorous, and the multitude of evaluated parameters is overwhelming. However, there are aspects that are not very clear or need to be modified.

The abstract: from my point of view, I would refrain from mentioning the working methods used, with details (stimulated 16 with LPS, then incubated for 24 hrs. in the presence or absence of EGCG).

Introduction is detailed and refers to all aspects of interest. I would still suggest some changes: there are repetitions of words/phrases (lines 54/55, 92 related to age-related), abbreviations Aβ (Amyloid-ß). Lines 68-70: Lipopolysaccharide (LPS), also known as endotoxin, is a gram-negative bacteria linked to many diseases, such as liver damage, neurological dysfunction, chronic inflammation of the gut, and diabetes(11).

Nothing is understood from this phrase. First of all, LPS are a structural part of the cell wall of Gram negative bacteria and have the function of endotoxin.

Also, there are available some new subject related references.

2. Materials and Methods

 2.1. Chemical and Reagents

Not all kits are mentioned (eg Pierce BCA Protein assay). Some materials are detailed in various chapters, others are not. The recommendation is to standardize the presentation. Considering that you used a lot of kits and reagents, you can opt for their synthetic mention in this chapter and a table for additional materials.

 2.4. Quantification of NO

Lines 147-149 should be reformulated

The next day cells were treated with control (no treatment), 150µM of EGCG only, 1µgm/L of LPS only, and 150µM of EGCG +1µg/ml of LPS, then incubated for 24 hrs)

 2.5. IL-6 and TNF-α ELISA Quantification

It is not understood how cells treated with different concentrations of EGCG were harvested: treated with EDTA trypsin for detachment or mechanical detachment.

The working protocol for the ELISA is presented almost in its entirety. From my point of view, it can be simplified by mentioning what type of ELISA it is, what detection range the standard scale allows and what the evaluation criteria were. If you maintain the entire protocol, specify which type of enzyme conjugate was used for each cytokine, the chromogenic substrate, the detection interval.

Also, the absorbance was read at 450, 540 and 570nm (all of them?)

 2.6. RNA Extraction and Reverse transcription to cDNA:

During and after washing step, were the cells centrifuged?

 (All centrifugation steps will be performed at >12,000 x g) - what speed was used for the study?

Lines 193-195: if you do the math with the amounts, you will obtain different results. Maybe it is a mistake with the RNA sample or the nuclease-free water.

 2.7. PrimePCR array Analysis of Inflammatory Cytokine/Chemokines, Akt3, mTOR, NF-κB, 198 and NO Signaling Pathways

 All the parameters that are investigated and that appear in the results part should be specified.

 lines 200-202 PrimePCR™ assay It is highly specific without using a probe, compatibility with standard assay conditions, avoided SNPs in target regions, a maximized fraction of transcript isoforms being detected, and utilizes the latest release of genome builds and annotation databases (reference?)

line 205 ul (µl)

 2.8. ProteinSimple Western Analysis

Lines 223-225: similar lines 147-149

In protein assay it is not mentioned if the standards and samples mixet with Pierce™ BCA Protein Assay reagent in a 96-well plate were incubated and in what condition.

 3. Results

3.3. EGCG attenuates pro-inflammatory cytokines

Mouse Quantikine ELISA Kit are tests that allows the quantitative assessment of IL-6 and TNF-a. However, in figures 3 and 4, the appreciation of the anti-inflammatory effect is made relative to the absorbance. What amount of cytokines was quantified?

 Line 292: 1µM LPS

 3.5. EGCG modulates neuroimmunomodulation of Inflammation via NF-κB, PI3k-Akt-mTOR, 333 and NO Pathways

The graphs showing the change in the expression of the investigated parameters must be detailed in full text. Some parameters appear in the figure, but they do not appear in the text. Some of them do not even appear in the additional materials (bcl3). I recommend checking the legends of the figures, where repetitions of parameters appear. RAF1: in the supplementary materials appears at PI3k-Akt elf4e (figure 9) does not appear in the text. In the supplementary material it is mentioned in the NO pathway as downregulated. GADDD45A???

 4. Discussion

Lines 467-470: We used LPS as the inflammatory inducer in which LPS was allowed to incubate for 1 hr. on the cells to create an inflammatory microenvironment, LPS coupled with EGCG 150µM was used to elicit an adaptive immune response, thus, to provide neuroprotection.

I don’t think that we can discuss about eliciting an adaptive immune response in vitro.

 Lines 519-521: Surprisingly, NF-kB evaluation displayed downregulation of IL-1b, which was not observed during the ELISA tests.

The evaluation by ELISA is not mentioned.

 I recommend using the same font to name a parameter (only capital letters for example) in text and figures.

It is necessary to standardize the name NF-κB, IL-1β

Author Response

Dear Editor

Attached is the revised version of manuscript ID: Brainsci-22242855 entitled Molecular Mechanisms of the Anti-inflammatory effects of Epigallocatechin 3 Gallate (EGCG) in Activated BV-2 Microglia Cells. We appreciate the reviewer's constructive evaluation, comments, and questions involving this work. We have addressed each of their concerns as follows:

Reviewer 1

The subject of this study is extremely interesting, considering that neurodegenerative diseases have a significant weight worldwide, and slowing down or stopping evolution is often impossible.

Anti-inflammatory effect of green Tea polyphenol Epigallocatechin-3-Gallate (EGCG) has been the subject of many studies, but the approach in this study is extremely complex. The methodology is extremely rigorous, and the multitude of evaluated parameters is overwhelming. However, there are aspects that are not very clear or need to be modified.

  1. The abstract: from my point of view, I would refrain from mentioning the working methods used, with details (stimulated 16 with LPS, then incubated for 24 hrs. in the presence or absence of EGCG).

  • Author's Response:

Lines 16-19 (originally lines 16-21) read as BV-2 microglia cells were grown, stimulated, and treated with EGCG. Cytotoxicity and nitric oxide (NO) production were evaluated. Immunoassay, PCR array, and WES™ Technology were utilized to evaluate inflammatory, neuroprotective modulators, and signaling pathways involved in the mechanistic action of neuroinflammation.

  1. The introduction is detailed and refers to all aspects of interest. I would still suggest some changes: there are repetitions of words/phrases (lines 54/55, 92 related to age-related), abbreviations Aβ (Amyloid-ß). Lines 68-70: Lipopolysaccharide (LPS), also known as endotoxin, is a gram-negative bacteria linked to many diseases, such as liver damage, neurological dysfunction, chronic inflammation of the gut, and diabetes (11). Nothing is understood from this phrase. First of all, LPS is a structural part of the cell wall of Gram-negative bacteria and has the function of endotoxin. Also, there are available some new subject-related references.

  • Author's response.
  • Lines 52-56 (originally lines 54-59) now reads, "They serve to maintain microglial homeostasis through the balance of noninflammatory cytokines, i.e., Interleukin 10 (IL-10, Interleukin 13 (IL-13), and pro-inflammatory cytokines such as Interleukin 6 (Il-6), Tumor necrosis factor alpha (TNF-α), Interleukin 1 beta (IL-1β), which are significantly expressed in AD."

  • Line 82-91 (originally lines 92-94) now reads, "Green tea has various medicinal properties to age-related neurodegeneration, such as antioxidative, anti-viral, anti-aging, and lipid-lowering activities (15, 16), due to its abundance of polyphenols or flavan-3-ols (also known as catechins).
  • Lines 67-71 (originally lines 68-72) Now read "Lipopolysaccharide (LPS), which is a glycolipid found in gram-negative bacteria, has been used as a microglial activator in various in vitro and in vivo model systems. (Batista et al., 2019; Bertani et al, 2018). Countless research papers have demonstrated LPS use due to its ability to mimic various inflammatory effects of cytokines/chemokines, such as TNF-α and Il-6, and its specific action on the TLR4 receptor (Farhana and Khan, 2022)." The introduction has been modified to include more current information related to neuroinflammation and associate it with the study's objectives and LPS.  
  •  
  1. Materials and Methods

2.1. Chemical and Reagents

Not all kits are mentioned (e.g., Pierce BCA Protein assay). Some materials are detailed in various chapters, while others are not. The recommendation is to standardize the presentation. Considering that you used a lot of kits and reagents, you can opt for their synthetic mention in this chapter and a table for additional materials.

  • Author's Response:

Chemicals and Reagents were revised.

Lines 104-124 (originally110-132) now read: "BV-2 microglial cells were kindly provided by Elizabeta Blasi (Blasi et al., 1990). Gibco DMEM, high glucose, GlutaMAX™ Supplement media, Gibco DMEM, high glucose HEPES, no phenol red, Gibco Penicillin-Streptomycin (10,000 U/mL), HBSS, and 10% heat-inactivated fetal bovine serum were purchased from Thermo Fisher Scientific (Waltham, MA). 50mg – (-) epigallocatechin Gallate  and Lipopolysaccharides (LPS) from Escherichia coli O111: B4 γ-irradiated were purchased from Sigma Aldrich (St Louis, MO, USA). Griess Reagent was purchased from Sigma Aldrich (St Louis, MO, USA). The individual ELISA kits for IL-6 and TNF-α were obtained from R&D Systems (Minneapolis, MN, USA). Aurum Total RNA mini kit, iScript™ Advanced cDNA Synthesis Kit,2-mercaptoethanol (14.2 M) 25 ml, ≥98% pure, SsoAdvanced™ Universal SYBR® Green Supermix, iScript™, Advanced cDNA Synthesis Kit, and the mouse PrimePCR™ inflammation array was purchased from Bio-Rad (Hercules, CA, USA). PrimePCR™ arrays utilized in this study were mTOR, NO, PI3K-Akt, and NF-B signaling pathways and were purchased from Bio-Rad (Hercules, CA, USA). Pierce BCA Protein Assay kit was purchased from Bio-Rad (Hercules, CA, USA). Protease and phosphatase inhibitors were obtained from G-Biosciences (Saint Louis, MO, USA). 12- 230 Separation Module, 8 x 25 capillary cartridges, Anti-Rabbit Detection Module   HeLa Lysate Controls, and Erk1 Primary Antibody were purchased from ProteinSimple (San Jose, CA USA). Primary Antibodies: NF-κB2 p100/p52(Akt3 (ElZ3W), mTOR (7C10) Rabbit mAb ) GAPDH  Alpha-actinin, and secondary antibody Anti-rabbit IgG-HRP were purchased from Cell Signaling (Danvers, MA, USA)."

  1. 4. Quantification of NO

Lines 147-149 should be reformulated. The next day cells were treated with control (no treatment), 150µM of EGCG only, 1µgm/L of LPS only, and 150µM of EGCG +1µg/ml of LPS, then incubated for 24 hrs.)

  • Author's Response:

Lines 145-154 (originally 144-154) have been changed to: "Briefly, BV-2 cells (5×104 cells/well, in a 96-well plate) were seeded overnight. The next day plates were prepared following the manufacturer's recommendations (Sigma Aldrich, St Louis, MO). An equal volume (50μL) of each experimental sample (cell culture supernatant), standard, and Griess Reagent were combined, incubated for 10 min at room temperature, and protected from light."

  1. 5. IL-6 and TNF-α ELISA Quantification

It is not understood how cells treated with different concentrations of EGCG were harvested: treated with EDTA trypsin for detachment or mechanical detachment. The working protocol for the ELISA is presented almost in its entirety. From my point of view, it can be simplified by mentioning what type of ELISA it is, what detection range the standard scale allows and what the evaluation criteria were. If you maintain the entire protocol, specify which type of enzyme conjugate was used for each cytokine, the chromogenic substrate, the detection interval.

Also, the absorbance was read at 450, 540, and 570nm (all of them?)

  • Author's Response:

Lines 157-164 (originally 157-163)  now read: "A total of 5×105 cells/mL (2mL per well) of BV-2 cells were seeded in 6-well plates overnight to attach. The next day cells were treated with control (no treatment), 150μM of EGCG only, 1µgm/L of LPS only, and 150μM of EGCG +1µg/ml of LPS, then incubated for 24 hrs. After 24hr exposure, the supernatant of each sample was collected into 5ml tubes and centrifuged at 1000 rpm for 5 min at 4˚C to remove any particulate material. IL-6 and TNF-α ELISA kits were used for quantitative measurement utilizing the manufacturer's instructions (R & D Systems, Minneapolis, MN).

Lines 172-178 (originally170-176) now read, "The absorbance was read at 450 and 540nm using the UV microplate spectrophotometer (model 7600, version 5.02, Cambridge Technologies Inc. (Worthington, MN, USA).  Following the manufacturer's recommendations: Optical errors were standardized by subtracting the absorbance of 540nm from 450nm. The standard curve was used to quantify the reading. Data generated from the Il-6/TNF-α Elisa kits were statistically quantified using GraphPad Prism 6 (version 6.07; Graph Pad Software Inc. San Diego, CA, USA by one-way ANOVA with Tukey's post hoc multiple comparisons test)

  1. .6. RNA Extraction and Reverse transcription to cDNA:

During and after the washing step, were the cells centrifuged? (All centrifugation steps will be performed at >12,000 x g) - what speed was used for the study?

Lines 193-195: if you do the math with the amounts, you will obtain different results. Maybe it is a mistake with the RNA sample or the nuclease-free water.

  • Author's Response

The centrifugation steps were conducted at 13,000 x g, following the manufacturer's protocol of greater than 12,000 x g. 

Lines 203-208 (originally lines 193-195) now read: "The cDNA strand was formulated from the RNA using Bio-Rad iScript advanced reverse transcriptase. Following the manufacturer's instructions, Briefly, A solution of 4µl of the 5X iScript advanced reaction mix, 1µl of reverse transcriptase, 5µl of the RNA sample (250ng/5µl) for RT-PCR assay, 9µl of nuclease-free water, and 1µl of the RT control assay template for a total of 20µl. The Reverse Transcription thermal cycling program included two steps: 46°C for 20 min and then 95°C for 5 min."

2.7. PrimePCR array Analysis of Inflammatory Cytokine/Chemokines, Akt3, mTOR, NF-κB, 198, and NO Signaling Pathways

All the parameters that are investigated and that appear in the results part should be specified.

 Lines 200-202 PrimePCR™ assay It is highly specific without using a probe, compatibility with standard assay conditions, avoided SNPs in target regions, a maximized fraction of transcript isoforms being detected, and utilizes the latest release of genome builds and annotation databases (reference?)

line 205 ul (µl)

  • Author's Response:

Lines 211-214 (originally lines 200-202) were rewritten as follows:
"PrimePCR™ assay assesses multiple genes related to a definite pathology. It is highly specific without using a probe, compatibility with standard assay conditions, avoided SNPs in target regions, a maximized fraction of transcript isoforms being detected, and utilizes the latest release of genome builds and annotation databases" was deleted.

  1. 8. ProteinSimple Western Analysis

Lines 223-225: similar lines 147-149

In protein assay, it is not mentioned if the standards and samples mixed with Pierce™ BCA Protein Assay reagent in a 96-well plate were incubated and in what condition.

  • Author's Response:
  • Lines 223-225, previously read: The following day, cells were treated with control (no treatment), 150μM of EGCG only, 1µg/mL of LPS only, and 150μM of EGCG +1µg/mL of LPS, then incubated for 24 hrs. Now reads: Cells were harvested as previously discussed, but the cell pellet was obtained and resuspended in 100µl of Lysis buffer (mixed with protease and phosphatase inhibitors), incubated on ice for 30 minutes. Lysate was sonicated for 3 seconds using a Sonic Dimembrator (Fisher Scientific, Hampton, NC). and centrifuged at 3,000RPM for 5 min. The supernatant (protein) was gathered, and the protein concentration was determined using the Pierce™ BCA Protein Assay Kit.

Lines 221-221 now read, "Cells were treated, harvested, and centrifuged as discussed above, and then the cell pellet was obtained and resuspended in 200µl of Lysis buffer (mixed with protease and phosphatase inhibitors), incubated on ice for 30 minutes. Lysate was sonicated for 30 seconds using a Sonic Dimembrator (Fisher Scientific, Hampton, NC). and centrifuged at 3,000RPM for 5 min. The supernatant (protein) was collected, and the protein concentration was determined using the Pierce™ BCA Protein Assay Kit. A series of con-centration standards ranging from 0 – 2 mg/mL were prepared using bovine serum albumin (BSA). 10µl of each unknown protein sample and standard were used (sample to working reagent ratio 1:20).200µl of the working reagent was added to each well of the 96 well microplates and mixed on a plate shaker for 30 secs. The plate was covered and incubated at 37°C for 30 minutes and left to cool at room temperature. Protein concentrations were quantified at 595 nm wavelength with a microplate reader Infinite M200 (Tecan Trading AG)."

  1. Results

3.3. EGCG attenuates pro-inflammatory cytokines.

Mouse Quantikine ELISA Kit are tests that allow the quantitative assessment of IL-6 and TNF-a. However, in figures 3 and 4, the appreciation of the anti-inflammatory effect is made relative to the absorbance. What amount of cytokines was quantified?

Line 292: 1µM LPS

  • Author's Response:
  • IL-6 and TNF-α ELISA kits were used for quantitative measurement utilizing the manufacturer's instructions (R & D Systems, Minneapolis, MN, USA). As described in the methods, The standards and samples were prepared according to the kit's directions. We used 100µl of each supernatant sample and Antigen Standard to conduct this experiment as described in the methods.
  • The figures for these experiments were modified. Figure 3 shows the amount of IL-6 reduced by EGCG 150µM+LPS was 350pg/ml when compared to LPS alone at 750pg/ml. Figure 4 displays EGCG 150µM+LPS increasing TNF-α by 30% or 1000pg/ml vs. LPS alone 700pg/ml.

  1. 5. EGCG modulates neuroimmunomodulation of Inflammation via NF-κB, PI3k-Akt-mTOR, 333, and NO Pathways

The graphs showing the change in the expression of the investigated parameters must be detailed in full text. Some parameters appear in the figure, but they do not appear in the text. Some of them do not even appear in the additional materials (bcl3). I recommend checking the legends of the figures, where repetitions of parameters appear. RAF1: in the supplementary materials appears at PI3k-Akt elf4e (figure 9) does not appear in the text. In the supplementary material, it is mentioned in the NO pathway as downregulated. GADDD45A???

  • Author's Response:
  • Lines 349-389 (originally 335-364) have been rewritten: "Inflammatory signaling pathways have been shown to become dysregulated and maladjusted, leading to destabilized cellular metabolism, defective clearance mechanisms, and heightened microglial stress response due to neurodegenerative disorder [30,31]. PrimePCR™ array evaluation of EGCG effects displayed the expression of many genes: Suppl. Table 2 lists in more detail all the genes that appeared in this analysis. Figures 7-14 show only the statistically significant genes for the aforementioned inflammatory signaling pathways. EGCG upregulated neuroprotective NF-κB genes (Figure 7), which were as follows: Colony stimulating factor 3 (CSF3), Toll-like receptor 4 (Tlr4), Toll-interleukin 1 receptor (TIR) domain-containing adaptor protein (TIRAP) Toll-like receptors (TLR1,3, and 4), Tumor necrosis factor (TNF), heme oxygenase 1(HMOX1), and ILR1 (Interleukin Receptor 1). EGCG downregulated pro-inflammatory NF-κB genes, as shown in Figure 8, which are: C-C-Ligand 5 (CCL5), Fos Proto-Oncogene, AP-1 Transcription Factor Subunit (FOS), Interleukin 1 Beta (IL1B), Mothers against decapentaplegic homolog 3 aka SMAD Family Member 3 (SMAD3), TNF receptor-associated factors (TRAF2.3, and 5), RelB Proto-Oncogene, NF-KB Subunit (RelB), Signal Transducer And Activator Of Transcription 1 (STAT1), V-RAF-leukemia viral oncogene 1 (RAF1), Nuclear Factor Kappa B p-100 subunit 2 (NF-κB2), and Mitogen-Activated Protein Kinase Kinase Kinase 1 (MAP3K1). EGCG upregulated immunosurveillance PI3K-AKT genes in Figure 9, which shows Eukaryotic Translation Initiation Factor 4E (ELF4E) and Toll-Like Receptor 4 (TLR4). In Figure 10, EGCG downregulated pro-inflammatory PI3K-AKT genes, which are as follows: Son of sevenless homolog 1 (Drosophila) (SOS1).Raf proto-oncogene serine/threonine-protein kinase aka proto-oncogene c-RAF (RAF1), Beta-Glucuronidase (GUSB), Insulin-Like Growth Factor I receptor (IGF1R), Cyclin-dependent kinase inhibitor 1B (CDKN1B), Growth Factor Receptor-Bound Protein 2 (GRB2), Mature T Cell Proliferation 1 (MTCP1), and Thymoma Viral Proto-Oncogene 3 (AKT3). Figure 11 displays EGCG downregulation of pro-inflammatory mTOR genes, which were: Regula-tory Associated Protein of mTOR Complex 1(RPTOR), Mitogen-Activated Protein kinase kinase (MAPK3), Thymoma Viral Proto-Oncogene 3 (AKT3). Protein Kinase Adenosine Monophosphate Activated Non-Catalytic Subunit 2 (PRKAB2), and mTOR. EGCG upregulated Insulin Signaling linked mTOR genes, as displayed in Figure 12, which are: Insulin2 (INS2) and Phospholipase D2 (PLD2). Figure 13 shows EGCG downregulation of oxidative stress-producing NO genes, which are: Glutathione Peroxidases 1 and 4 (GPX1 and 4), Growth Arrest and DNA Damage Inducible Protein (GADD45A), Nitric Oxide Synthase 1 (NOS1), Cathepsin B (CTSB), Cytochrome B-245 Alpha Chain (CYBA), Immunity-related GTPase family M protein aka Interferon-Inducible Protein 1 (IRGM1), Proliferation and Apoptosis Adaptor Protein 15A(PEA15A), and Hepsin (HPN). Finally, EGCG upregulated the bioenergetic regulatory GRB2-associated-binding protein 1 (Gab1). For this experiment, untreated cells were used as the negative control, and EGCG 150µM alone was utilized as the positive control. Variations of controls are due to the effects of EGCG on particular mRNA-expressed genes. Appendix B, Supplemental Table 2 gives further information related to each gene and fold change."
  • In response to the GADD45A downregulation, this could be due to an undocumented interaction with another protein (CDKN1B) or a signaling pathway (MAPK) that could interfere with regulating GADD4A, leading to a reduced expression. Another explanation may be that the stress-regulative role of GADD45A is being reduced to maintain cellular homeostasis or an immunoregulative response.

  1. Discussion

Lines 467-470: We used LPS as the inflammatory inducer in which LPS was allowed to incubate for 1 hr. on the cells to create an inflammatory microenvironment, LPS coupled with EGCG 150µM was used to elicit an adaptive immune response; thus, to provide neuroprotection.

I don't think that we can discuss about eliciting an adaptive immune response in vitro.

 Lines 519-521: Surprisingly, NF-kB evaluation displayed downregulation of IL-1b, which was not observed during the ELISA tests.

The evaluation by ELISA is not mentioned.

 I recommend using the same font to name a parameter (only capital letters, for example) in text and figures.

It is necessary to standardize the name NF-κB, IL1β.

  • Author's Response:
  • Lines 467-470 statement was deleted.
  • Lines 519-521 statement was deleted.

Reviewer 2 Report

Molecular Mechanisms of the Anti-inflammatory Effects of Epigallocatechin 3-Gallate (EGCG) in Activated BV-2 Microglia Cells

The authors focused the work in the anti-inflammatory effects of EGCG in activated BV-2 microglia cells. The aim of the research work was to determine the molecular targets underlying the anti-inflammatory effects of ECGC in BV-2 microglia cells after being stimulated by LPS. In order to carry out this work, the researchers determine cell viability, nitric oxide release, gene expression related to inflammatory signalling pathways and protein expression. The results presented in the manuscript are many, concretely, the data obtained by PrimePCR array Analysis of Inflammatory Cytokine/Chemokines, but the authors summarize the most relevant anti-inflammatory mechanisms in the discussion suggesting that EGCG could be a good candidate to avoid inflammation and consequently prevent neurodegenerative diseases.

The role of inflammation in neurodegeneration is becoming increasingly relevant, therefore, natural products such as EGCG, which can be found in green tea, are of great importance to prevent neurodegenerative diseases.

Even though the report is clear and the work provides interesting results, the manuscript has to be modified.

Comments and suggestions to improve the article are specified below.

Misspelling

-line 17: delete a dot after 24hrs

-line 271 add space after chose and 150

-line 278 add a space after 150

-line 522 change NFkβ by NFkB

Decide if you want to add a space after the concentrations or not. Follow the same in all manuscript.

Results 3.2

Figure2.

-In the legend of figure 2 is written in line 282 that: “The X-axis represents different concentrations of EGCG”. It is represented the different treatment conditions to BV-2 cells

-In line 283: “EGCG 150 (red) was identical” has to modify by: EGCG 150mM (red) was identical

-The authors specify the statistical differences in line 285, but they do not indicate versus which experimental condition. They should to modify the legend adding the comparisons.

Results 3.3

Figure3.

-What are you determining?

In material and methods, the authors described that: The absorbance was read at 450, 540, and 570nm using the UV microplate spectrophotometer (model 7600, version 5.02, Cambridge Technologies Inc. (Worthington, MN, USA). The standard curve was used to quantify the reading.

After reading the specifications in material and methods the cytokine release should be express in pg/ml?

-Legend of figure 3:

What does it mean positive control?? You should clarify why are you using a negative control, and which is your control. Rewrite the legend of the figure, also has to be explained what is reduced or increased versus which condition.

Results 3.4

-The authors have written in line 312: “Our results demonstrated that EGCG upregulated Interleukin 3 (IL-3), Interleukin 11 (IL-11), and Granulocyte-macrophage colony-stimulating factor (GM-CSF or CSF2) as shown in Figure 5A, B and C, respectively”

But EGCG alone does not cause any effect compare to control. The upreguladion is with LPS, it can cause a misunderstanding. The sentence should be modified by Our results demonstrated that the treatment of EGCG with LPS to BV-2 cells upregulated the expression of Interleukin 3 (IL-3), Interleukin 11 (IL-11), and Granulocyte-macrophage colony-stimulating factor (GM-CSF or CSF2).

Figure 5

-The bar scale of the graph should be the same for all the graphs represented in the same figure.

-Legend Figure 5 line 323: add upregulate after (purple) as is shown below:

“LPS (purple) upregulate the expression of IL-3 (A)”

-line 324: “A negative control is shown in blue, and positive control (red)”.

What does it mean negative and positive control?

It should be modified by: “A control is shown in blue, and EGCG in red”. (correct in all of the legends of the figures if it is necessary).

-line 325: it should be added the comparisons between the different experimental conditions as it has been mentioned above.

At the end of the legend it should be written: Values represent the mean ± S.D., *** p≤0.001 and ****p≤0.0001 vs LPS.

(correct in the rest of the figure legends when it is not specified in the legend text).

Figure 6

-The same suggestions that in figure 5.

Results 3.5

Figure 7

-In the legend of the figure 7, in line 373: “(Red)” should to be substitute by (Green).

-In line 374 at the end of the legend it should be written: Values represent the mean ± S.D., *p≤0.05, **p≤0.01, *** p≤0.001 373 and ****p≤0.0001. vs LPS.

(It should be correct in the rest of the figure legends if it is necessary).

Results 3.6

-In line 421: “EGCG promotes neuroprotection by diminishing protein expression of Akt3 and NF-κB2” it should be modified by:

EGCG promotes neuroprotection by diminishing protein levels of Akt3 and NF-κB2”

Figure 15

The authors shown in figure 15 A.  images of the protein levels of NF-κB2 and GAPDH. They should to provide the original images. In the manuscript the bands of the image are cropped and seems that they come from different western blots. Furthermore, the bands of GAPDH are not level out, indicating that these bands are not from the same western blot.

If the authors do not want to use the original western blot in the manuscript, then they should to specify at the legend of the figure that are showing representative images of the NF-κB2 and GAPDH levels from three independent experiments. Although, in that case, the image should to be improved by one with better resolution.

(It should be done in all of the figures where is shown representative images of protein levels).

Figure 16

-Legend of figure 16, line 446: add ***p≤0.001, ****p≤0.0001

Figure 17

-Legend of figure 17, line 440: delete ***p≤0.001

Discussion

The authors analysed a large number of genes that are altered with LPS treatment in BV-2 cells, which were modified by the action of EGCG. The data to be analysed are many and in the discussion is not easy to relate all of them.

The genes chosen by the researchers to determine protein levels were indicative of EGCG effects on the signalling pathways themselves, as NF-κB2, mTOR, and Akt3, the results obtained by treating BV-2 cells with EGCG after LPS treatment are very interesting. But the authors, in the discussion (lines 543 to 549) do not argue much about these results, and only cite a single reference in each case. They should expand the argument on how EGCG treatment is able to significantly decrease the levels of these 3 proteins activated by LPS and their relation with microglial anti-inflammatory effects supported by more references.

Figure 18

The summary represented in Figure 18, displays the mechanism by which EGCG decreases microglial activation in a model of LPS. Accordingly, it should be highlighted in the legend that these effects are caused through microglial activation since the work has been carried out in BV-2 microglial cells.

Conclusion

In the conclusion the authors used a summary of the results instead of to emphasize the relevant data that is extracted from the work, that is the effect of EGCG on activated microglia.

The authors can delete the fragment from line 567 to 575. They should point out the last paragraph on the effects of EGCG, and specify that this work was done in microglial cells, therefore more studies should be done either in vitro and in vivo to clear up the anti-inflammatory effects of EGCG.

Author Response

Molecular Mechanisms of the Anti-inflammatory Effects of Epigallocatechin 3-Gallate (EGCG) in Activated BV-2 Microglia Cells

The authors focused the work on the anti-inflammatory effects of EGCG in activated BV-2 microglia cells. The aim of the research work was to determine the molecular targets underlying the anti-inflammatory effects of ECGC in BV-2 microglia cells after being stimulated by LPS. In order to carry out this work, the researchers determine cell viability, nitric oxide release, gene expression related to inflammatory signaling pathways, and protein expression. The results presented in the manuscript are many, concretely, the data obtained by PrimePCR array Analysis of Inflammatory Cytokine/Chemokines, but the authors summarize the most relevant anti-inflammatory mechanisms in the discussion suggesting that EGCG could be a good candidate to avoid inflammation and consequently prevent neurodegenerative diseases.

The role of inflammation in neurodegeneration is becoming increasingly relevant; therefore, natural products such as EGCG, which can be found in green tea, are of great importance in preventing neurodegenerative diseases.

Even though the report is clear, and the work provides interesting results, the manuscript has to be modified.

Comments and suggestions to improve the article are specified below.

  1. Misspelling

-line 17: delete a dot after 24hrs

-line 271, add space after choice and 150

-line 278, add a space after 150

-line 522 change NFkβ by NFkB

Decide if you want to add a space after the concentrations or not. Follow the same in all manuscript.

  • Author's Response:
  • Spelling errors have been corrected.
  • Lines 16-19 (originally lines 16-17) have been changed to read: "BV-2 microglia cells were grown, stimulated, and treated with EGCG. Cytotoxicity and nitric oxide (NO) production were evaluated. Immunoassay, PCR array, and WES™ Technology were utilized to evaluate inflammatory, neuroprotective modulators, and signaling pathways involved in mechanistic action of neuroinflammation."
  • Line 271 now, line 287 now reads
    "EGCG 150µM was chosen as the concentration for the rest of the experiments."
  • Line 278, now line 273, has been altered.
  • Line 522, now line 530 NF-κB, has also been corrected to NF-κB
  1. Results 3.2

Figure 2.

-In the legend of figure 2 is written in line 282 that: "The X-axis represents different concentrations of EGCG." It is represented the different treatment conditions for BV-2 cells.

-In line 283: "EGCG 150 (red) was identical" has to modify by EGCG 150mM (red) was identical

-The authors specify the statistical differences in line 285, but they do not indicate versus which experimental condition. They should modify the legend by adding the comparisons.

  • Author's response:

  • Lines 284-290 (originally 282-286) were changed to "The X-axis represents the different treatment conditions to BV-2 cells, while the Y-axis shows NO production (%).EGCG 150µM (red) showed no significant change compared to the control (blue). EGCG 150µM paired with 1µg/ml of LPS (purple) significantly reduced NO generation compared to LPS (green). LPS(1µg/ml) showed a 45% increase in NO production when compared to the controls (EGCG 150µM alone and no treatment control), whereas the EGCG 150µM with 1µg/ml LPS displayed a 25% decrease in NO generation. Values represent the mean ± SD, *** p≤0.001 and ****p≤0.0001 vs. LPS."

  1. Results 3.3

    Figure3.

-What are you determining?

In material and methods, the authors described that: The absorbance was read at 450, 540, and 570nm using the UV microplate spectrophotometer (model 7600, version 5.02, Cambridge Technologies Inc. (Worthington, MN, USA). The standard curve was used to quantify the reading.

After reading the specifications in material and methods, the cytokine release should be expressed in pg/ml?

-Legend of figure 3:

What does positive control mean?? You should clarify why you are using a negative control and which is your control. Rewrite the legend of the figure; also, it must be explained what is reduced or increased versus which condition.

  • Author's Response:

Lines 170-176 have been modified. The graphs for Figures 3 and 4 have been corrected and put in pg/ml.

The controls have been defined in the text. The phrases positive control and negative control have been deleted. Our experiment treatment conditions include Control (Untreated cells), EGCG 150µM alone, LPS (1µg/ml), and   EGCG 150µM with 1µg/ml LPS as shown above in the figures.

The legend of Figure 3 now reads: 1µg/ml LPS (green) showed a 45% increase in IL-6 production when compared to both controls (Untreated cells and EGCG 150µM alone) (Colors shown in blue and red, respectively). EGCG 150µM with 1µg/ml LPS showed a 25% decrease in IL-6 production when compared with LPS (1µg/ml). Values represent the mean ± SD, ***p≤0.001, and ****p≤0.0001 vs. LPS.

  1. Results 3.4

-The authors have written in line 312: "Our results demonstrated that EGCG upregulated Interleukin 3 (IL-3), Interleukin 11 (IL-11), and Granulocyte-macrophage colony-stimulating factor (GM-CSF or CSF2) as shown in Figure 5A, B and C, respectively".

But EGCG alone does not cause any effect compared to control. The upregulation is with LPS; it can cause a misunderstanding. The sentence should be modified by "Our results demonstrated that the treatment of EGCG with LPS to BV-2 cells upregulated the expression of Interleukin 3 (IL-3), Interleukin 11 (IL-11), and Granulocyte-macrophage colony-stimulating factor (GM-CSF or CSF2). 

  • Author's Response
  • Lines 320-322 (originally lines 312-314) now read, "Our results demonstrated that EGCG 150µM coupled with 1µg/ml LPS elevated the gene expression of Interleukin 3 (IL-3), Interleukin 11 (IL-11), and Granulocyte-macrophage colony-stimulating factor (GM-CSF or CSF2) as shown in Figure 5."

  1. Figure 5

-The bar scale of the graph should be the same for all the graphs represented in the same figure.

-Legend Figure 5 line 323: add upregulate after (purple) as is shown below:

"LPS (purple) upregulates the expression of IL-3 (A)"

-line 324: "A negative control is shown in the blue, and positive control (red)."

What does it mean negative and positive control?

It should be modified by: "A control is shown in blue, and EGCG in red." (correct in all of the figures' legends if necessary).

-line 325: it should be added the comparisons between the different experimental conditions as it has been mentioned above.

At the end of the legend, it should be written Values represent the mean ± SD, *** p≤0.001, and ****p≤0.0001 vs. LPS.

(correct the rest of the figure legends when it is not specified in the legend text).

Figure 6

-The same suggestions that in figure 5.

  • Author's Response:

Figure 5. The bar scale is not the same because the gene expression of the cytokines was different for each based on the EGCG effect. Each gene was normalized to GAPDH using the Bio-Rad CFX96 Manager software.

  • The legend of Figure 5 Lines 338-338 (originally 322-325) has been revised to read "PCR array analysis of BV-2 cells display EGCG 150µM+ LPS (1µg/ml) (purple) significantly increased mRNA expression of IL-3 (A), IL-11 (B), and CSF2 (C) compared to LPS(1µg/ml) (green). LPS (1µg/ml) showed a non-significant increase when compared to EGCG 150µM with LPS (1µg/ml). The controls (denoted in blue and red, respectively) displayed reduced mRNA expression when compared to LPS and EGCG 150µM with LPS(1µ/ml). Values represent the mean ± SD, *** p≤0.001, and ****p≤0.0001 vs. LPS.
  • The legend of Figure 6 lines 341-346 (originally lines 328-332) now reads: PCR array evaluation of BV-2 cells demonstrated that EGCG 150µM +LPS (purple) reduces the mRNA expression of pro-inflammatory mediators: MIF (A), CCL2 (B), and TNFS10 (C). LPS 1µg/mL (green) considerably elevated mRNA expression of MIF, CCL2, and TNFS10 when contrasted to 150µM and 1µg/ml LPS. The controls (denoted in blue and red, respectively) varied in expression due to the effects of EGCG on these specific genes. Values represent the mean ± SD, *** p≤0.001, and ****p≤0.0001 vs. LPS.
  1. Results 3.5

Figure 7

-In the legend of figure 7, in line 373: "(Red)" should be substituted by (Green).

-In line 374 at the end of the legend, it should be written: Values represent the mean ± SD, *p≤0.05, **p≤0.01, *** p≤0.001 373, and ****p≤0.0001. vs. LPS.

(It should be correct in the rest of the figure legends if it is necessary).

  • Author's Response:

Lines 393-458 (originally lines 369-446) Figure 7-14 have been changed to read as follows: Controls are shown in blue and red, respectively. Values represent the mean ± S.D., *p≤0.05, **p≤0.01, *** p≤0.001 373 and ****p≤0.0001. vs. LPS.

  1. Results 3.6

-In line 421: "EGCG promotes neuroprotection by diminishing protein expression of Akt3 and NF-κB2," it should be modified by:

"EGCG promotes neuroprotection by diminishing protein levels of Akt3 and NF-κB2."

  • Author's Response

Figure 15

The authors showed in figure 15 A.  images of the protein levels of NF-κB2 and GAPDH. They should provide the original images. In the manuscript, the bands of the image are cropped, and it seems that they come from different western blots. Furthermore, the bands of GAPDH are not level out, indicating that these bands are not from the same western blot.

If the authors do not want to use the original western blot in the manuscript, then they should specify at the legend of the figure that shows representative images of the NF-κB2 and GAPDH levels from three independent experiments. Although, in that case, the image should be improved by one with better resolution.

(It should be done in all of the figures where is shown representative images of protein levels).

Figure 16

-Legend of figure 16, line 446: add ***p≤0.001, ****p≤0.0001

Figure 17

-Legend of figure 17, line 440: delete ***p≤0.001

  • Author's Response:

The originals have been added as supplemental material (see Supplemental Figure 1. NF-κB2 protein expression normalizing with GAPDH. Supplemental Figure 2. AKT3 protein expression normalizing with GAPDH. Supplemental Figure 3. mTOR protein expression normalizing with α-Actinin.). As suggested, the legends of Figure 15-17 reads Western Blot representative image of NF-κB2, AKT3, and mTOR, respectively.
Figure 16 and 17 legends have been modified as suggested.

  1. Discussion

The authors analyzed a large number of genes that are altered with LPS treatment in BV-2 cells, which were modified by the action of EGCG. The data to be analyzed are many and in the discussion is not easy to relate all of them.

The genes chosen by the researchers to determine protein levels were indicative of EGCG effects on the signaling pathways themselves, as NF-κB2, mTOR, and Akt3, the results obtained by treating BV-2 cells with EGCG after LPS treatment are very interesting. But the authors, in the discussion (lines 543 to 549) do not argue much about these results and only cite a single reference in each case. They should expand the argument on how EGCG treatment is able to significantly decrease the levels of these 3 proteins activated by LPS and their relationship with microglial anti-inflammatory effects supported by more references.

Figure 18

The summary, represented in Figure 18, displays the mechanism by which EGCG decreases microglial activation in a model of LPS. Accordingly, it should be highlighted in the legend that these effects are caused by microglial activation since the work has been carried out in BV-2 microglial cells.

  • Author's Response
  • Lines 551-585 (originally lines 543-554) have been modified to address the genes that were evaluated for western blot analysis NF-κB2, mTOR, and AKT3. It reads as follows: "The genes chosen for Protein Simple WES™ evaluation were indicative of EGCG effects on the signaling pathways themselves, i.e., NF-κB2, mTOR, and Akt3. The inhibition of NF-κB2 by EGCG may elicit further study on the effects of NF-κB in promoting inflammation and its role in mediating aging [72]. AKT3 is one of the isoforms of the PI3K-Akt signaling pathway; it is found within the neurons and comprises 50% of the total mammalian brain [73]. This AKT isoform has a dearth of research related to microglia and neurodegeneration. A role may exist in macrophage regulation [74] Polytarchou et al.,2020 [75] demonstrated that AKT3 could generate oxidative stress and DNA breakdown by stimulating the NADPH oxidase through the phosphorylation of p47phox using an in vitro murine model system. Further investigation of AKT3 is its ability to modulate mitochondria and autophagy [76,77]. Most importantly, AKT3 has been shown to be involved in lysosomal dysregulation caused by cellular senescence [78]. Dubois et al.,2019 [73] showed that AKT3 modulates protection against demyelinating inflammatory disorder. Nutraceutical intervention of the PI3K-AKT pathway in microglial regulation may be promising to consider the shortage of scientific investigation [79]. EGCG downregulation of AKT3 may utilize an anti-aging function that mediates AKT3 activity.
  • mTOR is a multifaceted kinase that regulates autophagy, cellular senescence, aging, and neuroinflammation [80-82]. mTOR also is involved in autophagy regulation [83]. More importantly, mTOR is a possible treatment for AD pathogenesis [17]. EGCG displayed a reduction of mTOR in western blot analysis. EGCG action, similar to other nutraceuticals, may be beneficial to mitochondrial bioenergetics via mTOR signaling intervention [84,85].
  • NF-κB2 is the gene that encodes for the NF-κB family of proteins. NF-κB is an extensively studied inflammatory pathway. As previously mentioned, NF-κB regulates NO, which mediates oxidative stress and microglial activation via TLR signaling [86]. Some less studied elements of NF-κB action correlated to AD are aging control [87], Estrogenic regulative action [88], and Flavonoid neuroprotection [89]. WES Technology showed that EGCG downregulated NF-κB, which shows a flavonoid intervention in regulating inflammatory action. More research is necessary to understand the roles of mTOR, NF-kB, and Akt signaling in mediating the lipid metabolic effects on aging and neuroinflammation [90,91]. Research mechanisms of Tau protein contributions to aggregation were previously discussed in our prior work [92], but their relationship to microglial inflammation [93] and signaling mechanisms has yet to be determined. Our research shows that EGCG may be able to quell microglial inflammation and act as an anti-aging preventive measure in early-stage neurodegeneration."
  • Figure 18 has been modified to read: "Schematic diagram showing a possible mechanism of EGCG inhibition of LPS-induced inflammatory and oxidative stress response leading to depression of neuroinflammation. Experimentation showed that EGCG diminished LPS-activated BV-2 microglial cells via the following suggested mechanisms: EGCG halted the NF-κB and the PI3K-AKTpathway, which downregulates mTORC1, thus causing an increase in its anti-inflammatory and neuroprotective attributes. EGCG also controls Insulin signaling to mediate PI3k-AKT. EGCG directs the regulation of nitric oxide via NF-κB signaling, which results in the reduction of oxidative stress/metabolic stress. Also, transcriptional modulators may be involved in further neuro-rescue and anti-senescent capabilities of EGCG."

  1. Conclusion

In the conclusion, the authors used a summary of the results instead of emphasizing the relevant data that is extracted from the work, that is, the effect of EGCG on activated microglia.

The authors can delete the fragment from lines 567 to 575. They should point out the last paragraph on the effects of EGCG and specify that this work was done in microglial cells; therefore, more studies should be done either in vitro or in vivo to clear up the anti-inflammatory effects of EGCG.

  • Author's Response

The conclusion has been replaced to now discuss the major findings and novelty of this study as well as to point out possible directions for the future.

  • Lines 598-605 (originally 567-580) now reads:

"Our research showed that EGCG shows a profound mechanism of regulating PI3K-AKT, mTOR, and NO signaling in order to diminish neuroinflammation. EGCG's anti-inflammatory and neuroprotective attributes were displayed by its ability to lessen the action of well-established inflammatory, stimulating cytokines, i.e., IL-6 and MIF. The neuroprotective capabilities were exhibited by the upregulation of IL-3 and IL-11. EGCG displayed some autophagic properties by acting on CTSB and PRKAB2. Although LPS was utilized to stimulate the BV-2 microglial cells, EGCG demonstrated the ability to suppress microglial initiation, which is a pivotal event in neurodegenerative disease.

Reviewer 3 Report

The manuscript analyzed the anti-inflammatory and antioxidant effects of EGCG on microglia. However, there're a series of problems lacking innovation and normal form in the manuscript, which make the present manuscript is questionable. There are many reports about anti-inflammation and antioxidation of EGCG in vitro, and no further improvement is found in this crude manuscript. Thus, it is necessary for the authors to add more information to make the novelty and significance of this work clearer, as well as to clearly explain what concrete mechanism behind, and how the EGCG strengthen that, in preventing neurodegenerative diseases. My comments are listed as below.

– Check the style of the unit writing, such as the unit “L”.

– Putting a blank space in front of unit.

– Spelling mistake of NF-κB, it is not the ‘NF-κβ’.

– It is suggested for the authors to perform more experiments on the mechanism analysis, as mentioned above.

– The results of ProteinSimple Western Analysis don't look normal, please provide a full picture of original data.

Author Response

Reviewer 3

The manuscript analyzed the anti-inflammatory and antioxidant effects of EGCG on microglia. However, they're a series of problems lacking innovation and normal form in the manuscript, making the present manuscript questionable. There are many reports about the anti-inflammation and antioxidation of EGCG in vitro, and no further improvement is found in this crude manuscript. Thus, it is necessary for the authors to add more information to make the novelty and significance of this work clearer, as well as to clearly explain what concrete mechanism is behind and how the EGCG strengthens in preventing neurodegenerative diseases. My comments are listed below.

– Check the style of the unit writing, such as the unit "L."

– Putting a blank space in front of the unit.

– Spelling mistake of NF-κB; it is not the 'NF-κβ.'

– It is suggested that the authors perform more experiments on the mechanism analysis, as mentioned above.

– The results of ProteinSimple Western Analysis don't look normal; please provide a full picture of the original data.

  • Author's Response:
  • Yes, there have been numerous reports about the anti-inflammation and antioxidative properties of EGCG in vitro. Nevertheless, our research was innovative in that EGCG revealed downregulated genes and proteins in signaling pathways, including mTOR, PI3K-AKT, NO, and NF-κB, which are implicated in neuroinflammation and oxidative stress. mTOR regulates autophagy and neuroinflammation (Thakur et al.,2023). The downregulation of mTOR by EGCG enhanced its anti-inflammatory and neuroprotective properties. In microglia, PI3K-AKT is important in regulating neuroinflammatory processes (Cianciulli et al., 2020). EGCG was able to inhibit PI3K-AKT, which in turn controls mTOR signaling pathways. NF-KB also regulates oxidative stress and neuroinflammation (Wang et al., 2022). EGCG was able to inhibit NF-κB2 and related genes involved in its regulation (RELB) and cytokine expression (IL-6, CCL2, and MIF). The downregulation of NF-κB and associated genes by EGCG demonstrates a role for flavonoids in regulating inflammatory activity. It has been demonstrated that nitric oxide (NO) plays a role in neuroinflammation and oxidative stress related to neurodegeneration (Liy et al.,2021). EGCG reduced NO production and genes related to the NO signaling pathway (GPX1, GPX4, and NOS1), which leads to a reduction in oxidative stress and neuroinflammation. EGCG's ability to act indirectly on the Insulin signaling pathway is linked to causing inflamed microglia due to aging (Haas et al., 2020). In summary, a proposed mechanism of EGCG suppression of the inflammatory and oxidative stress response caused by LPS is shown in Figure 18.

Figure 18. Schematic diagram showing a possible mechanism of EGCG inhibition of LPS-induced inflammatory and oxidative stress response leading to depression of neuroinflammation. Experimentation showed that EGCG diminished LPS-activated BV-2 microglial cells via the following suggested mechanisms: EGCG halted the NF-κB and the PI3K-AKTpathway, which downregulates mTORC1, thus causing an increase in its anti-inflammatory and neuroprotective attributes. EGCG also controls Insulin signaling to mediate PI3k-AKT. EGCG directs the regulation of nitric oxide via NF-κB signaling, which results in the reduction of oxidative stress/metabolic stress. Also, transcriptional modulators may be involved in further neuro-rescue and anti-senescent capabilities of EGCG.

References used for Reviewer 3 response.

  1. Thakur, S., Dhapola, R., Sarma, P., Medhi, B., & Reddy, D. H. (2023). Neuroinflammation in Alzheimer's Disease: Current Progress in Molecular Signaling and Therapeutics. Inflammation, 46(1), 1–17. https://doi.org/10.1007/s10753-022-01721-1
  2. Wang, C., Fan, L., Khawaja, R. R., Liu, B., Zhan, L., Kodama, L., Chin, M., Li, Y., Le, D., Zhou, Y., Condello, C., Grinberg, L. T., Seeley, W. W., Miller, B. L., Mok, S. A., Gestwicki, J. E., Cuervo, A. M., Luo, W., & Gan, L. (2022). Microglial NF-κB drives tau spreading and toxicity in a mouse model of tauopathy. Nature communications, 13(1), 1969. https://doi.org/10.1038/s41467-022-29552-6
  3. Cianciulli, A., Porro, C., Calvello, R., Trotta, T., Lofrumento, D. D., & Panaro, M. A. (2020). Microglia Mediated Neuroinflammation: Focus on PI3K Modulation. Biomolecules, 10(1), 137. https://doi.org/10.3390/biom10010137
  4. Liy, P. M., Puzi, N. N. A., Jose, S., & Vidyadaran, S. (2021). Nitric oxide modulation in neuroinflammation and the role of mesenchymal stem cells. Experimental biology and medicine (Maywood, NJ), 246(22), 2399–2406.
  5. Haas, C. B., de Carvalho, A. K., Muller, A. P., Eggen, B. J. L., & Portela, L. V. (2020). Insulin activates microglia and increases COX-2/IL-1β expression in young but not in aged hippocampus. Brain Research, 1741, 146884.

  • Sorry about the unit "L" and putting the space before the unit is not clear for me.
  • NF-κβ was corrected to NF-κB
  • Please see the above response regarding the proposed mechanism of EGCG inhibition of LPS-induced inflammatory and oxidative stress response.

Figures 15-17 show the representative images of western bands for NF-κB2, AKT3, and mTOR, respectively, from three independent experiments. The originals have been added as supplemental material (see Appendix C-E) Supplemental Figure 1. NF-κB2 protein expression normalizing with GAPDH. Supplemental Figure 2. AKT3 protein expression normalizing with GAPDH. Supplemental Figure 3. mTOR protein expression normalizing with α-Actinin.).

Round 2

Reviewer 1 Report

The authors partially answered the questions and made changes. For example, the detection range for cytokines is not specified.

Author Response

Reviewer 1

Comment: The authors partially answered the questions and made changes. For example, the detection range for cytokines is not specified.

Author's Response:

Lines 162-185 have been rewritten to the following:

This experiment used R & D Systems, mouse IL-6, and TNF-α ELISA kit to measure IL-6 and TNF-α release into BV-2 cell supernatant quantitatively. A total of 5×105 cells/mL of BV-2 cells were seeded in 6 well plates (2mL per well) overnight to attach. The following day cells were treated with control (no treatment), 150μM of EGCG only, 1µgm/L of LPS only, and 150μM of EGCG +1µg/ml of LPS, then incubated for 24 hrs. After 24hrs exposure, the supernatant of each sample was collected into 5ml tubes and centrifuged at 1000 rpm for 5 min at 4˚C to remove any particulate material. IL-6 and TNF-α ELISA kit reagents include the following: 96 well plate, the wash buffers, assay diluents, color reagents A &B, stop solution, conjugates, and standards. Briefly, seven serial diluted standards in pg/ml (700, 350, 175, 87.5, 43.8, 21.9, 10.9) and blank (no standard) were prepared. 100µl (in triplicate) of the standard, blank, and supernatant samples were added to each well in the plate, covered with an adhesive strip, then incubated for 2 hr. at room temperature. Following this, the plate was washed, then 100µl of TNF-α or IL-6 conjugate was added to each well and incubated at room temperature for 2 hr. The plate was rewashed, and 100µl of substrate solution was added to each well and incubated in the dark for 30 minutes. Finally, 100µl of stop solution was added, then the optical density of each well was read at 450 and 540nm using the microplate reader (model 7600, version 5.02, Cambridge Technologies Inc. (Worthington, MN, USA). Optical errors were calculated by subtracting the absorbance of 540nm from 450nm. The standard curve was created by plotting the average of the triplicate optical density of each of the seven serial diluted standards. The concentration(pg/ml) of the samples was determined using the standard curve. This data obtained (concentrations of the sample[pg/ml]) was analyzed using GraphPad Prism 6 (version 6.07; Graph Pad Software Inc. San Diego, CA, USA by one-way ANOVA with Tukey's post hoc multiple comparisons test).

Reviewer 3 Report

The results of ProteinSimple Western Analysis don't look normal.

Author Response

Reviewer 3

Comment: The results of ProteinSimple Western Analysis do not look normal.

Author’s Response

Yes, the bands obtained from the ProteinSimple (WES) analysis may look different from the traditional western blot methods. This is due to the SimpleWestern™ operates by using a capillary system. The prepared plate is loaded into a capillary electrophoresis machine, and the proteins are separated by size as they move through a stacking and separation matrix. The separated proteins are then covalently attached to the capillary well through proprietary photoactivated capture chemistry (no use in transferring protein to a membrane). In this method,  the target proteins are recognized using a primary antibody and determined using a primary antibody chemiluminescent substrate or a fluorescently labeled secondary antibody. The bands obtained from the WES are more rectangular and precise. The resulting signal is identified and quantified using Compass for Simple Western soft
